# mTOR inhibition in Q175 Huntington's disease model mice facilitates neuronal autophagy and mutant huntingtin clearance

**Philip Stavrides[1], Chris N Goulbourne[1], James Peddy[1], Chunfeng Huo[1], Mala Rao[1,2], Vinod Khetarpal[3], Deanna M Marchionini[3], Ralph A Nixon[1,2,4,5]\*, Dun-Sheng Yang[1,2]\***

[1]Center for Dementia Research, Nathan S. Kline Institute, Orangeburg, United States; [2]Department of Psychiatry, New York University Grossman School of Medicine, New York, United States; [3]CHDI Management/CHDI Foundation, New York, United States; [4]Neuroscience Institute, New York University Grossman School of Medicine, New York, United States; [5]Department of Cell Biology, New York University Grossman School of Medicine, New York, United States

**\*For correspondence:**
Ralph.Nixon@nki.rfmh.org (RAN);
Dun-Sheng.Yang@nki.rfmh.org (D-SY)

**Competing interest:** The authors declare that no competing interests exist.

## eLife Assessment

This study presents an **important** finding on the alterations in the autophagic-lysosomal pathway in a Huntington's disease model. The evidence supporting the claims of the authors is **convincing**. The original reviewers have found most of the issues previously raised have been addressed although further suggestions are given for consideration. These comments are listed below. The work will be of interest to neuroscientists working on HD.

**Abstract** Huntington's disease (HD) is caused by the expansion of the polyglutamine stretch in huntingtin protein (HTT) resulting in hallmark aggresomes/inclusion bodies (IBs) composed of mutant huntingtin protein (mHTT) and its fragments. Stimulating autophagy to enhance mHTT clearance is considered a potential therapeutic strategy for HD. Our recent evaluation of the autophagic-lysosomal pathway (ALP) in human HD brain reveals upregulated lysosomal biogenesis and relatively normal autophagy flux in early Vonsattel grade brains, but impaired autolysosome clearance in late grade brains, suggesting that autophagy stimulation could have therapeutic benefits as an early clinical intervention. Here, we tested this hypothesis by crossing the Q175 HD knock-in model with our autophagy reporter mouse TRGL (**T**hy-1-**R**FP-**G**FP-**L**C3) to investigate *in vivo* neuronal ALP dynamics. In the Q175 and/or TRGL/Q175 mice, mHTT was detected in autophagic vacuoles and also exhibited a high level of colocalization with autophagy receptors p62/SQSTM1 and ubiquitin in the IBs. Compared to the robust lysosomal pathology in late-stage human HD striatum, ALP alterations in Q175 models are also late-onset but milder, that included a lowered phospho-p70S6K level, lysosome depletion, and autolysosome elevation including more poorly acidified autolysosomes and larger-sized lipofuscin granules, reflecting impaired autophagic flux. Administration of a mTOR inhibitor to 6-mo-old TRGL/Q175 normalized lysosome number, ameliorated aggresome pathology while reducing mHTT-, p62-, and ubiquitin-immunoreactivities, suggesting the beneficial potential of autophagy modulation at early stages of disease progression.

## Introduction

HD is an autosomal dominant disorder caused by a mutation in the gene encoding HTT resulting in expansion of the polyglutamine (polyQ) stretch on its amino-terminus (*Bates et al., 2015*; *Franklin et al., 2024*; *Jiang et al., 2023*; *Macdonald, 1993*). HD pathogenesis advances in a spatiotemporal pattern, which is used to stage disease pathology severity as Grade 0–4 (HD0, HD1, HD2, etc.) (*Vonsattel et al., 1985*). GABA-containing medium spiny projection neurons in the striatum are most susceptible to cell death (*Vonsattel, 2007*), and the cerebral cortex, particularly layer 5 a, also shows cell loss (*Pressl et al., 2024*; *Sotrel et al., 1991*). Neuronal intranuclear inclusions (NIIs) and neuropil inclusions are present in HD brains and are positive for mHTT and ubiquitin (Ub) (*DiFiglia et al., 1997*; *Gutekunst et al., 1999*).

Recently, we completed a comprehensive evaluation of the ALP in the HD brain at progressive disease stages, focusing on the most affected brain region, the striatum, in comparison to a less affected region, the neocortex (*Berg et al., 2024*). Double fluorescence immunolabeling and immuno-electron microscopy (IEM) revealed colocalization of HTT/mHTT with the autophagy-related adaptor proteins, p62/SQSTM1 and ubiquitin, and cathepsin D (CTSD) within aggresome inclusions and autophagic compartments, documenting the involvement of ALP in HTT/mHTT turnover and the disease-related impairment of this process in late-stage disease. The temporal evolution of ALP alterations generally revealed minimal detectable impairment of upstream autophagy steps [e.g., autophagosome (AP) induction, formation, and fusion with lysosomes (LY)] and, in striatum, elevated levels of LAMP1 and LAMP2 markers suggesting modestly upregulated LY biogenesis. At late disease stages, mainly HD4, neuronal ALP dysfunction exhibited enlarged/clumped CTSD-immunoreactive autolysosomes (AL)/LY and ultrastructural evidence of autophagic vacuole (AV) fusion and transition to lipofuscin granule formation. These findings collectively suggest that relatively competent autophagy machinery is maintained during the disease progression with a compensatory upregulation in lysosomal biogenesis, which together prevents against mHTT accumulation. This situation is failing at the late disease stages when AL clearance is impeded, substrates, including mHTT and its metabolites, accumulate in AL, and aggresome inclusions increase.

A possible implication of the foregoing findings is that pharmacologic enhancement of autophagy applied at a symptomatic but early stage of disease, when the ALP clearance machinery is fully competent, may be therapeutic in clearing mHTT protein in affected neurons. By contrast, in Alzheimer's Disease (AD) lysosomal clearance deficits develop at the earliest disease stages, suggesting enhanced autophagy induction is counterproductive.

Modulation of autophagy as a therapeutic strategy for HD has been investigated in various cell and animal models of HD (*Boland et al., 2018*; *Jiang et al., 2023*; *Kim et al., 2021*; *Sarkar and Rubinsztein, 2008b*; *Yang and Zhang, 2023*). Of particular relevance to our present study are those involving autophagy induction with mTOR-dependent or -independent autophagy-enhancing approaches in HD mouse models [e.g. Trehalose *Tanaka et al., 2004*; Rapamycin analog Temsirolimus (CCI-779) *Ravikumar et al., 2004*; Rilmenidine *Rose et al., 2010*; Rhes manipulations *Baiamonte et al., 2013*; *Lee et al., 2015*]. These studies have generally demonstrated ameliorative effects on outcome measures such as mHTT lowering and behavioral/motor function assays. However, in many cases, it is unclear whether autophagy mechanisms are directly engaged in the brain or are critical to rescue.

Thus, our study aimed to comprehensively characterize the autophagy response with a range of autophagy markers to interrogate the competence of the entire autophagy process in relationship to mHTT in the well-characterized zQ175 Knock-In HD mouse model (Q175). Towards this goal, we evaluated autophagy in neurons of Q175 after introducing by neuron-specific transgenesis the dual fluorescence-tagged autophagy probe, **t**andem **f**luorescent mRFP-eGFP-**LC3** (tfLC3), an autophagy adaptor protein associated with AP and degraded via autophagy (*Lee et al., 2019*). We thereby generated a new Q175 cross, namely TRGL (**T**hy-1-**R**FP-**G**FP-**L**C3)/Q175. tfLC3 expression, driven postnatally by the neuron-specific Thy1-promoter, allows for selective monitoring of neuronal autophagy without the confounding influence of glial cells. Resolution and sensitivity in reporting tfLC3 signal is high compared to conventional immunofluorescence staining of LC3. The ability to ratiometrically report pH-dependent changes in fluorescence (hue angle) enables neutral pH AP to be distinguished from acidic AL, that progressively acidify intraluminally upon fusion with LY. We are further able to differentiate AL subgroups differing in their extent of acidification after LY fusion. Distinguishing properly acidified AL from those poorly acidified because of delayed or defective acidification is assisted

by immunolabeling AL with a LY marker, such as CTSD, which is then detected with a third fluorophore. Together, this triple fluorescence paradigm is objectively quantified by computer-assisted deconvolution of the proportions of each label within the analyzed neuron, a reflection of their relative pH (and fusion with LY), which allows ALP organelle subtypes including LC3-negative LY to be identified and quantified for their numbers, sizes and spatial distributions in intact brain sections. The collective data provides reports on the completion (or lack thereof) of autophagy flux (ALP dynamics) and any blockage at particular steps in the ALP pathway, including the normal acidification and further maturation of AL through their successful elimination of fluorescence-tagged LC3 (*Lee et al., 2019*; *Lee et al., 2022*; *Lie et al., 2021*; *Lie et al., 2022*).

Crossing TRGL mice with a model of a neurodegenerative disease, the Q175 mouse model of HD, has enabled us to assess disease-related autophagy alterations in the Q175 and TRGL/Q175 models and their response to a pharmacological inhibition of mTOR, (mTORi) INK-128 (hereafter INK). Our data demonstrate target engagement and positive effects of the compound on rescuing Q175 phenotypes including reversal on AL/LY subtypes as reported by the tfLC3 probe and parallel reductions of mHTT-, p62- and Ub-immunoreactivity (IR), suggesting that the compound targeted the ALP to degrade mHTT.

## Results

### Identification of inclusions and HTT molecular species in Q175 mice

Immunohistochemistry (IHC) experiments with the antibody mEM48, which preferentially recognizes aggregated mHTT (*Gutekunst et al., 1999*), revealed age-dependent development of mHTT-positive profiles in the striatum of Q175KI mice. Brain sections from 2.5-mo-old Q175 (not shown) only exhibited faint and diffuse nuclear mHTT staining without identifiable aggresomes/inclusion bodies (IBs), while at 6 and 10 mo of age, mHTT-positive IBs were readily detected progressively with age (*Figure 1A2–A3*). There were no similar mHTT immunoreactive puncta in WT striatum (*Figure 1A1*). These observations are consistent with previous findings in Q175 and other HD mouse models (*Carty et al., 2015*; *Li et al., 1999*; *Menalled et al., 2003*). To further determine the locations of mHTT IBs in Q175, mEM48-immunostained sections were counterstained with cresyl violet (*Figure 1A4*) to distinguish nuclear IBs (*Figure 1A4*, arrowheads) from extranuclear IBs, which were localized predominantly in the neuropil (*Figure 1A4*, arrows) and detected, but rarely, in the cytoplasmic portion of the perikaryon. Thus, our results demonstrate an age-dependent increase in mHTT aggresomes in the Q175 model.

To reveal the ultrastructural locations and features of the mHTT aggregates, immunogold EM (IEM) with antibody mEM48 was performed. EM images (*Figure 1B*) demonstrated that the IEM with this antibody was highly specific in detecting mHTT IBs which were localized in the neuronal nuclei, dendrites, and axons. Ultrastructurally, most IBs were cotton-ball shaped and composed of fine fibrous or granular elements, somewhat similar to the unbundled short fibrils/protofibrils found *in vitro* with recombinant mHTT protein fragments (*Ko et al., 2018*; *Kolla et al., 2021*; *Mario Isas et al., 2021*), and similar to the structure of NII type inclusions we found in the human HD brain (*Berg et al., 2024*). Notably, however, human neuritic IBs often displayed a more heterogeneous composition, such as fine fibrils mixed with AVs or bundles composed of microtubule-like filaments, which were not found in the mouse neuritic IBs. Thus, the result suggests a homogeneous aggresome pathology in the mouse model, possibly reflecting a more rapid formation rate than in humans.

Previous immunoblotting studies have observed fragmentation of mHTT molecules in the human brain (*Kim et al., 2001*; *Mende-Mueller et al., 2001*), including our own study which detects mHTT fragments of 45–48 kDa, which predominantly exist in HD striatum (*Berg et al., 2024*). To assess HTT molecular species in the Q175 mouse brain, we employed multiple antibodies that preferentially detected mHTT over wild-type HTT, including MAB1574 (Clone 1C2) (*Figure 1C1 and C2*), its epitope containing a 38-glns stretch (*Trottier et al., 1995*), and mAb PHP2 (*Figure 1C1*), reacting with the peptide sequence QAQPLLPQP within the proline-rich domain of HTT (*Ko et al., 2018*). Both detected full-length mHTT and a~120 kDa fragment in the Q175 model (*Figure 1C1*; 1C2 left and 1C3 top two graphs). By contrast, MAB5490 (*Figure 1C2* right), reacting with aa115-129 of HTT (C-terminal to the region of polyQ stretch-containing exon 1), detected both wild type and mutant forms of HTT (*Figure 1C3*, bottom right graph). A~48 kDa HTT fragment may correspond to the

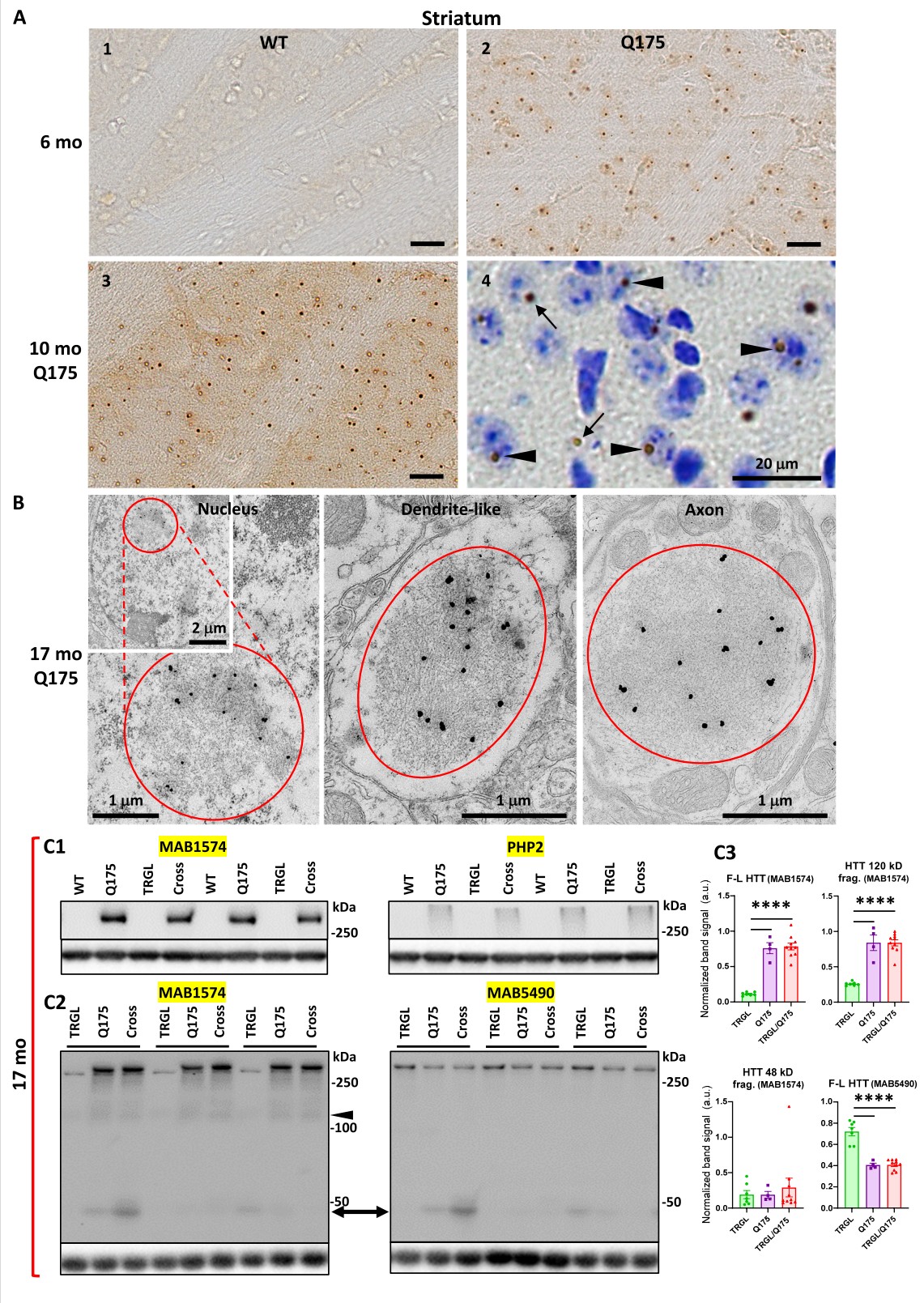

**Figure 1.** Identification of inclusions and huntingtin protein (HTT) molecular species in Q175 mice. (**A**) Immunohistochemistry (IHC) detects age-dependent increase in the number of mutant huntingtin protein (mHTT) inclusions. Brain sections from 6-mo- and 10-mo-old wild-type (WT) and Q175 mice were processed for IHC with antibody mEM48 (MAB5374) directed against mHTT (**A**). Images from sections without (**A1-3**) or with (**A4**) a cresyl violet counterstain are shown where the dark-brown puncta represent mHTT-positive inclusions. Arrowheads depict neuronal intranuclear inclusions

*Figure 1 continued on next page*

*Figure 1 continued*

(NIIs), determined with the assistance from the nuclear labeling by cresyl violet, while arrows indicate extranuclear inclusions, primarily the neuritic inclusion in the neuropil. Bars = 20 μm. n=4 mice/genotype, 4 sections/mouse. (**B**) mHTT inclusions are detected in nucleus, dendrites, and axons of Q175 brains by immuno-gold electron microscopy (IEM). Sagittal vibratome brain sections of 17-mo-old Q175 were cut and went through electron microscope (EM) processing. Small blocks were obtained from the striatal areas for ultrathin sectioning. Tissue containing grids were processed for immunogold labeling procedure with antibody mEM48, using 10 nm gold followed by silver enhancement. Structures showing high level of silver-enhanced gold labeling were considered as mHTT-positive. (**C**) Various forms of HTT molecules are detected with different antibodies by immunoblotting. Equal amounts of proteins from hemibrain homogenates of 17-mo-old WT, TRGL, Q175, and TRGL/Q175 (labeled as 'Cross') were subjected to SDS-PAGE and processed for WB with different antibodies directed against HTT/mHTT, including MAB1574 (**C1, C2**), mAb PHP2 (**C1**) and MAB5490 (**C2**). Images were collected by a digital gel imager (Syngene G:Box XX9). The arrowhead and arrow (**C2**) depict a 120 kDa and a 48 kDa fragment, respectively. (**C3**) Densitometry was performed with Image J for the blots shown in (**C2**) and the results were normalized by the immunoblot(s) of given loading control protein(s) (e.g., GAPDH). Values are the Mean ± SEM for each group (n=7 TRGL, 4 Q175, and 10 TRGL/Q175). Significant differences among the groups were analyzed by one-way ANOVA followed by Sidak's multiple comparisons test. *$p<0.05$, **$p<0.01$.

The online version of this article includes the following source data for figure 1:

**Source data 1.** Original western blots for *Figure 1C1 and C2*.

**Source data 2.** Original western blots for *Figure 1C1 and C2*, labeled for the relevant bands.

45–48 kDa fragment seen in human brain (*Berg et al., 2024*), and was detected by both MAB1574 and MAB5490 antibodies in some samples (*Figure 1C2*) but its levels in the Q175 models and the control TRGL were not statistically significantly different (*Figure 1C3*, bottom left graph). Thus, the result suggests that HTT fragmentation is not obvious in the Q175 brains, unlike the far more prevalent occurrence of this phenomenon in human HD striatum.

## mHTT colocalizes with p62, Ub, and CTSD in Q175 striatum

Our earlier study in human HD brains *Berg et al., 2024* found a high degree of colocalization of mHTT/Ub or Ub/p62 colocalized signals in IBs, suggesting a relationship between mHTT and the autophagy machinery since p62 and Ub are adaptor proteins mediating autophagic cargo sequestration. Similarly, in Q175 mice, mHTT/Ub or Ub/p62 signals were highly colocalized in IBs, particularly in NIIs, in striatal neurons of Q175 mice (*Figure 2—figure supplement 1*). Additional triple IF labeling experiments (mHTT/Ub/p62) with brain sections from 6- and 10-mo-old Q175 mice identified IBs positive for mHTT, p62, and Ub within or outside the nuclei (*Figure 2A*, arrows and arrowheads, respectively). The 10-mo-old Q175 mice exhibited more and larger mHTT-positive IBs than 6-mo mutants (*Figure 2A*, first column), consistent with *Figure 1A* and the literature (*Carty et al., 2015*; *Deng et al., 2021*). Thus, our data demonstrate a close spatial relationship among mHTT and autophagy receptor protein p62 and Ub in the Q175 model, which is consistent with our observations in human HD brain (*Berg et al., 2024*).

Double IF labeling of mHTT with a LY marker CTSD (*Figure 2B1*) detected small punctate mHTT signal in CTSD-positive vesicles, suggesting a pool of mHTT within the ALP. Consistent with this LM finding, IEM with antibody MAB1574 directed against mHTT clearly demonstrated that the AVs in either cell bodies, dendrites, and axons were labeled with concentrated silver-enhanced gold particles (*Figure 2B2*), and the specificity of the IEM labeling was very high, as reflected by a much higher number of silver-enhanced gold particles associated with AVs versus the minimal number of silver-enhanced gold particles with mitochondria that represent the background labeling (*Figure 2B3*). Together, the LM and IEM findings suggest that mHTT molecules exist within ALP vesicles in the absence of definable aggresome/IB structures, similar to the IEM finding from a recent report (*Zhou et al., 2021*).

## Mild late-onset alterations in the ALP revealed in 17-mo-old Q175 mice

We crossed TRGL mice (*Lee et al., 2019*) with Q175 to generate TRGL/Q175 and assessed AV/LY subtypes in striatal neurons with a hue angle-based analysis method (*Lee et al., 2019*; *Lee et al., 2022*). Our pilot studies in young TRGL/Q175 mice (2.5–6-mo of age) hardly detected autophagy alterations, compared to the control TRGL mice (not shown). Previous studies have reported that autophagy impairment was hardly detected in 6-mo Q175/GFP-LC3 mice (*Wold et al., 2016*), but alterations in a limited number of autophagy markers (e.g. protein levels of p62, LC3, p-Beclin-1) were found in brain homogenates of 12–15-mo Q175 mice (*Abd-Elrahman et al., 2017*; *Heikkinen et al.,*

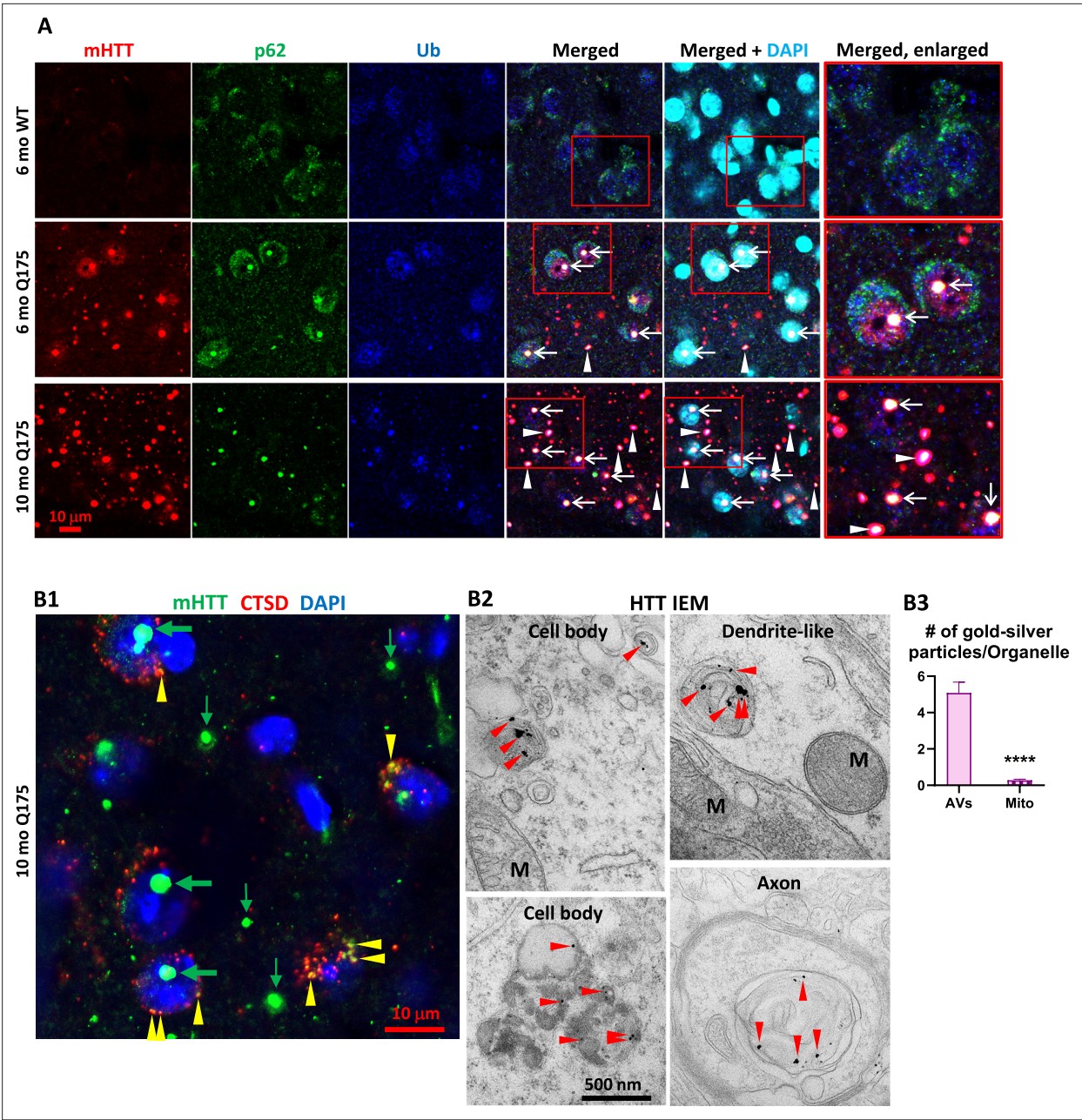

**Figure 2.** Colocalization of mutant huntingtin protein (mHTT) with p62, Ub, and cathepsin D (CTSD). (**A**) Triple labeling detects colocalization of mHTT with both autophagy adaptor proteins p62 and Ub. Brain sections from 6-mo- and 10-mo-old mice were immunostained with antibodies to huntingtin protein (HTT) (antibody MW8; red), p62 (green), or pan-Ub (blue), followed by an additional DAPI (cyan) labeling, and confocal images from the striatum are shown. Boxed areas are enlarged and shown in the last column. Arrows and arrowheads depict the white areas representing triply labeled inclusions containing the signals of the three proteins, where arrows are for NIIs, determined with the assistance from the nuclear DAPI labeling, while the arrowheads are for extranuclear inclusions. Bar = 10 µm. n=4 mice/genotype, 4 sections/mouse. (**B**) mHTT is detected in vesicles of the autophagic-lysosomal pathway (ALP). (**B1**) Brain sections from 10-mo-old Q175 were double-immunolabeled with antibodies to HTT (antibody MW8; green) and CTSD (red), followed by an additional DAPI (blue) labeling, and a three-color-merged confocal image from the striatum is shown. Large and small green arrows depict HTT nuclear and extranuclear inclusions, respectively, while yellow arrowheads depict yellow puncta showing HTT and CTSD signal colocalization. Bar = 10 µm. n=4 mice/genotype, 4 sections/mouse. (**B2**) Immuno-gold electron microscopy (IEM) with anti-HTT antibody (MAB1574) specifically detects HTT signal, represented by the silver-enhanced gold particles (red arrowheads), in autophagic vacuoles (AV)/lysosomes (LY) in cell bodies, dendrites, and axons. (**B3**) To demonstrate the labeling specificity of the HTT antibody in this IEM study, the number of silver-enhanced gold particles in AV/LY existing in neuronal cell bodies and neurites was counted from 69 electron microscope (EM) images from two 10-mo-old Q175 mice against the number of silver-enhanced gold particles in mitochondria on the same images, and the result is shown in the bar graph. Statistical significances between the two groups were analyzed by unpaired, two-tailed t-test. ****p<0.0001.

*Figure 2 continued on next page*

*Figure 2 continued*

The online version of this article includes the following figure supplement(s) for figure 2:

**Figure supplement 1.** Colocalization of huntingtin protein (HTT)/Ub and Ub/p62 in IBs in the striatum (STR).

*2021*; *Wold et al., 2016*). Therefore, to further validate the status of the ALP and to discover potential additional autophagy alterations, we expanded our study to old mice at 17-mo of age, where we found the following ALP alterations.

## Alterations in AV/LY subtypes

Confocal images from 17-mo-old TRGL/Q175 and control TRGL brain sections, triple-fluorescent labeled via an additional immunostaining with an anti-CTSD antibody and far-red emitting fluorophore, showed enhanced mRFP- and eGFP-LC3 signals in the striatal neurons of TRGL/Q175 compared to TRGL (*Figure 3A1*). Hue angle-based analysis confirmed the above observations by revealing significant increases in the numbers of AL and poorly-acidified AL (pa-AL, as explained in *Lee et al., 2019*; *Lee et al., 2022*). It also detected a reduction in the number of LY (*Figure 3A2*). Thus, these results indicate that the tfLC3 reporter can reveal alterations in the proportions of AV/LY subtypes in Q175, consistent with delayed and/or deficient acidification of AL causing deficits in the reformation of LY to replenish the LY pool.

## Mild alterations in the levels of autophagy marker proteins detected by immunoblotting

To further assess autophagy phenotypes in Q175 models, we conducted western blotting (WB) using hemibrain homogenates from 17-mo-old mice for protein markers of individual steps in the ALP (i.e. autophagy induction, membrane nucleation/AP formation, autophagy adaptor proteins and AL formation/substrate degradation) (*Figure 3—figure supplements 1–4*). At the stage of autophagy induction signaling, we did not see alterations in the levels of mTOR and p-mTOR forms including its auto-phospho-form at S2481, but we did observe a decreased level of a mTOR substrate, p-p70S6K (T389), in TRGL/Q175 compared to the control TRGL (*Figure 3—figure supplement 1*), implying reduced mTOR activity, similar to the finding from a previous study (*Ravikumar et al., 2004*). Another mTOR substrate, p-ULK1 (S757), did not exhibit changes in TRGL/Q175. However, there was an increase in the level of total ULK1, resulting in a reduced ratio of p-ULK1 (S757)/total ULK1 in TRGL/Q175 compared to TRGL (*Figure 3—figure supplement 1*). The data collectively imply a down-regulated mTOR activity, as expected in response to accumulated aggregate-prone protein in Q175 mice, or as a result of sequestration of mTOR by mHTT IBs, as suggested previously (*Ravikumar et al., 2004*).

Except for these alterations, we did not detect statistically significant differences between TRGL/Q175 and the control TRGL in all other tested marker proteins in the downstream phases of the ALP (*Figure 3—figure supplements 2-4*). Of special interest is the ATG14-containing VPS34/Beclin-1 complex implicated in HD pathogenesis (*Park et al., 2016*; *Wold et al., 2016*) for which we did not detect statistically significant alterations in the levels of Beclin-1, VPS34, and ATG14 and their corresponding phosphor-forms. Notably, even if some TRGL/Q175 clearly exhibited diminished signals for p-ATG14 (S29) (*Figure 3—figure supplement 2*), no statistical significance could be established due to large variations among samples (See Discussion). Another notable point is that we did not find alterations in levels of full-length proteins or fragments (*Jamilloux et al., 2018*; *Norman et al., 2010*; *Sanchez-Garrido et al., 2018*; *Valionyte et al., 2022*) of autophagy adaptor proteins such as p62, TRAF6 (*Figure 3—figure supplement 3*), in contrast to our immunoblot analysis of human HD brains (*Berg et al., 2024*). Together, our data suggest that, in general, autophagy alterations at the protein level, as can be detected by immunoblotting, are mild in TRGL/Q175 even at 17-mo-old (See Discussion).

## Increased numbers of larger lipofuscin granules

To assess possible ultrastructural pathologies, we conducted an EM study for Q175 vs WT at 17-mo of age. As predicted, NIIs were not observed in WT neurons (*Figure 3B*, left panel) but were detected (*Figure 3B*, right panel, red circle) in ~10% striatal neurons of Q175 mice. Different from the substantial accumulation of AVs including lipofuscin granules in late stage human HD brain (e.g., HD4) (*Berg*

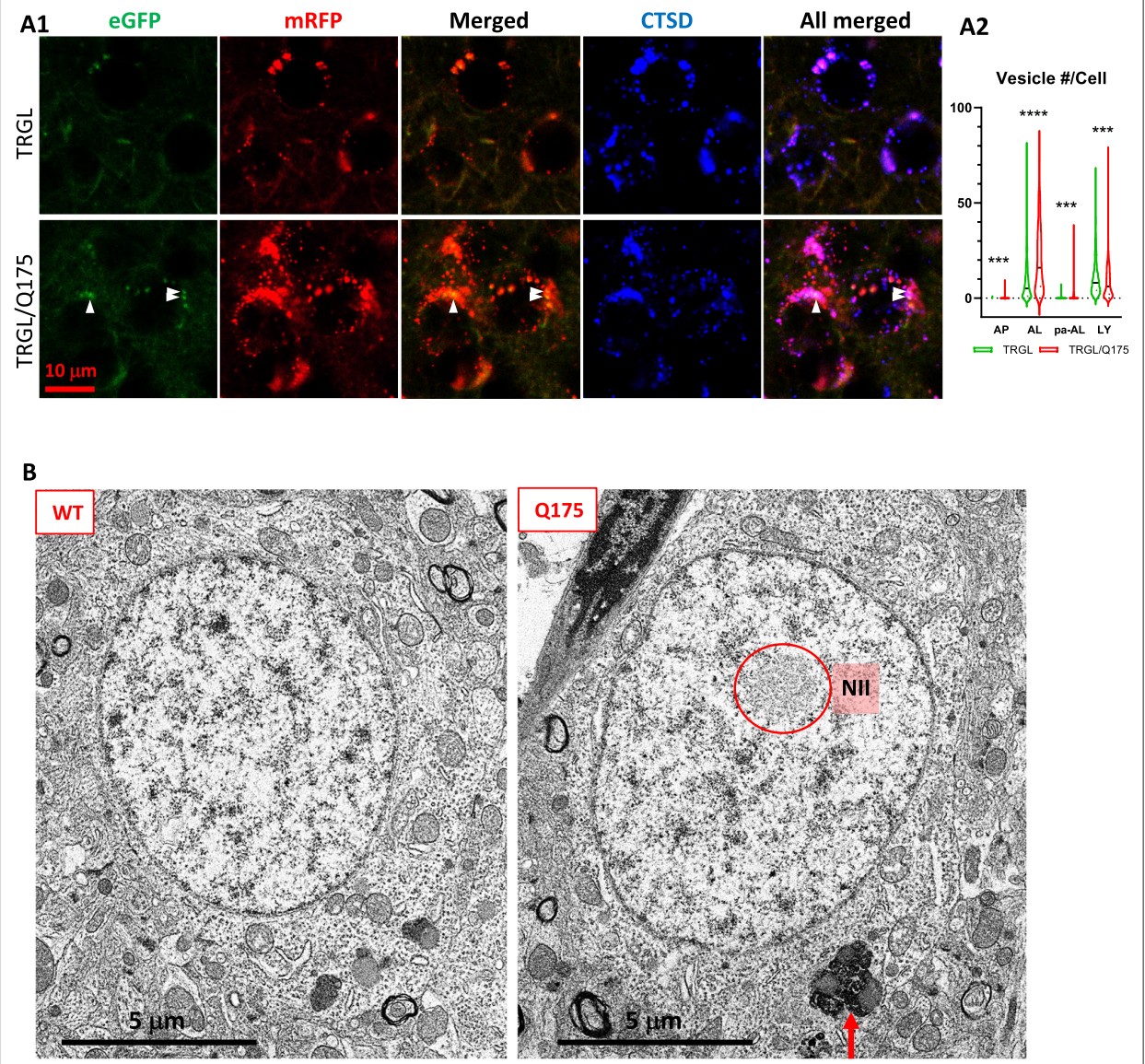

**Figure 3.** Mild late-onset alterations in the autophagic-lysosomal pathway (ALP) in the striatum of 17-mo-old Q175. (**A**) Quantitation of autophagic vacuoles (AV)/lysosomes (LY) subtypes of striatal neurons detect increases in AL, pa-AL, and a decrease in LY in 17-mo-old TRGL/Q175 vs. TRGL. (**A1**) Brain sections from TRGL and TRGL/Q175 (4 sections/mouse, 10 mice/genotype) were immunostained with an anti-CTSD antibody. Confocal images from the cranial-dorsal portion of the striatum (three images at 120 x/section) were collected and representative images for each eGFP-LC3 (green), mRFP-LC3 (red), and CTSD (blue) are shown. Arrowheads depict pa-AL. (**A2**) Hue angle-based analysis was performed for AV/LY subtype determination using the methods described in *Lee et al., 2019* (see the Materials and methods). Data are presented as Vesicle #/Neuron (TRGL: n=713 neurons; TRGL/Q175: n=601 neurons). Statistical significances between the two groups for each vesicle type were analyzed by unpaired t-test. Two-tailed p-value: ***p<0.001, ****p<0.0001. (**B**) EM detects larger AL/lipofuscin granules in the Q175 striatum. Sagittal vibratome brain sections from 17-mo-old mice were cut and went through EM processing. Small blocks were obtained from the striatal area for ultrathin sectioning, followed by EM examinations of the grids. The red circle depicts NII, and the arrow depicts larger sized (>1 μm) lipofuscin granules, which were counted on randomly collected images of neurons from striatum of WT (420 neurons from n=6 mice) or Q175 (586 neurons from n=9 mice).

The online version of this article includes the following source data and figure supplement(s) for figure 3:

**Figure supplement 1.** Molecules involved in autophagy induction signaling of the autophagic-lysosomal pathway (ALP) are largely unchanged in 17-mo-old TRGL/Q175.

**Figure supplement 1—source data 1.** Original western blots for *Figure 3—figure supplement 1*.

**Figure supplement 1—source data 2.** Original western blots for *Figure 3—figure supplement 1*, labeled for the relevant bands.

*Figure 3 continued on next page*

*Figure 3 continued*

**Figure supplement 2.** Molecules involved in membrane nucleation/autophagosome (AP) formation of the autophagic-lysosomal pathway (ALP) are largely unchanged in 17-mo-old TRGL/Q175.

**Figure supplement 2—source data 1.** Original western blots for *Figure 3—figure supplement 2*.

**Figure supplement 2—source data 2.** Original western blots for *Figure 3—figure supplement 2*, labeled for the relevant bands.

**Figure supplement 3.** Autophagy adaptor proteins in the autophagic-lysosomal pathway (ALP) are largely unchanged in 17-mo-old TRGL/Q175.

**Figure supplement 3—source data 1.** Original western blots for *Figure 3—figure supplement 3*.

**Figure supplement 3—source data 2.** Original western blots for *Figure 3—figure supplement 3*, labeled for the relevant bands.

**Figure supplement 4.** Molecules involved in autolysosomes (AL) formation/substrate degradation of the autophagic-lysosomal pathway (ALP) are largely unchanged in 17-mo-old TRGL/Q175.

**Figure supplement 4—source data 1.** Original western blots for *Figure 3—figure supplement 4*.

**Figure supplement 4—source data 2.** Original western blots for *Figure 3—figure supplement 4*, labeled for the relevant bands.

*et al., 2024*), we did not observe gross accumulation of AVs in Q175 at this old age, except that quantitative analysis of EM images revealed that larger sized (>1 μm) lipofuscin granules (*Figure 3B*, right panel, arrow) modestly increased in number in Q175 compared to WT [WT: 113 lipofuscin/420 neurons (n=6 mice), i.e., 27 lipofuscin/100 neurons; Q175: 228 lipofuscin/586 neurons (n=9 mice), i.e., 39 lipofuscin/100 neurons; Unpaired t-test, Two-tailed p-value <0.01; Bar graph not shown]. Thus, these EM findings suggest that ultrastructural autophagy alterations in the ALP were mild in Q175 even at an older age.

## DARPP-32-IR decreases in striatal neurons of 17-mo-old TRGL/Q175 in the absence of neuronal loss

To investigate potential neurodegeneration in our model, brain sections from 17-mo-old mice were double-labeled with the medium spiny neuron marker DARPP-32 (dopamine- and cAMP-regulated phosphoprotein, 32 kDa)(*Ouimet et al., 1998*) and a general neuronal marker NeuN. We found that the intensity of DARPP-32-IR significantly diminished in TRGL/Q175 compared to TRGL (*Figure 4A and B*). However, there were no alterations in NeuN-IR, including Area Covered and the Number of NeuN-positive neurons (*Figure 4A and B*), and there were clear examples of neurons showing strong NeuN signal but faint or no DARPP-32 signal (*Figure 4A*, arrows). Such a finding is consistent with other studies in Q175 and Q140 mice where neuronal loss was not found even though a reduction in DARPP-32 was seen in the same study, and that neuronal loss only occurred quite late, e.g., around 2 y of age (*Deng et al., 2021*; *Hickey et al., 2008*; *Peng et al., 2016*; *Rothe et al., 2015*). Thus, our data support the notion that changes in DARPP-32 may indicate alterations in its protein level and a loss of phenotype pattern, which is a potentially common prelude to neuronal loss in other neuronal cell types, e.g., basal forebrain cholinergic neurons (*Jiang et al., 2022*).

## Oral mTORi INK engages mTOR target and induces downstream responses in the ALP in 7-mo-old TRGL/Q175

As a prelude to investigate effects of pharmacological autophagy stimulation with a mTORi, INK, on TRGL/Q175 mice, we first performed a safety, pharmacokinetic, and pharmacodynamic evaluation of INK in 6-mo-old WT mice, administered via oral gavage. The WB results (*Figure 5—figure supplement 1A*) indicated that all doses tested, from 2.5 mg/kg (mpk) to 10-mpk, exhibited target engagement as revealed by the reductions in the protein levels of mTOR targets, i.e., p-ULK1 (S757, phosphorylated by mTOR), p-p70S6K (not shown), or phosphorylated S6 ribosomal protein (p-S6, at S240/244), implying high BBB permeability of this compound and target engagement. Consistently, measurements for the levels of INK in cerebellar homogenates of WT mice treated with 1, 3, and 10-mpk INK revealed dose-dependent brain levels of INK (*Figure 5—figure supplement 1B*).

We then decided on an oral gavage treatment regimen for INK as 4-mpk, daily, for 3 wk, to be administered in 6-mo-old TRGL/Q175. The brains from the mice (7-mo of age after the 3 wk treatment duration) were then analyzed by multiple experimental approaches. By immunoblotting, target engagement was verified, as indicated by decreased levels of p-mTOR (S2481, autophosphorylation site), p-ULK1 (S757) and p-S6 (S240) in brain homogenates from INK-treated TRGL/Q175 compared

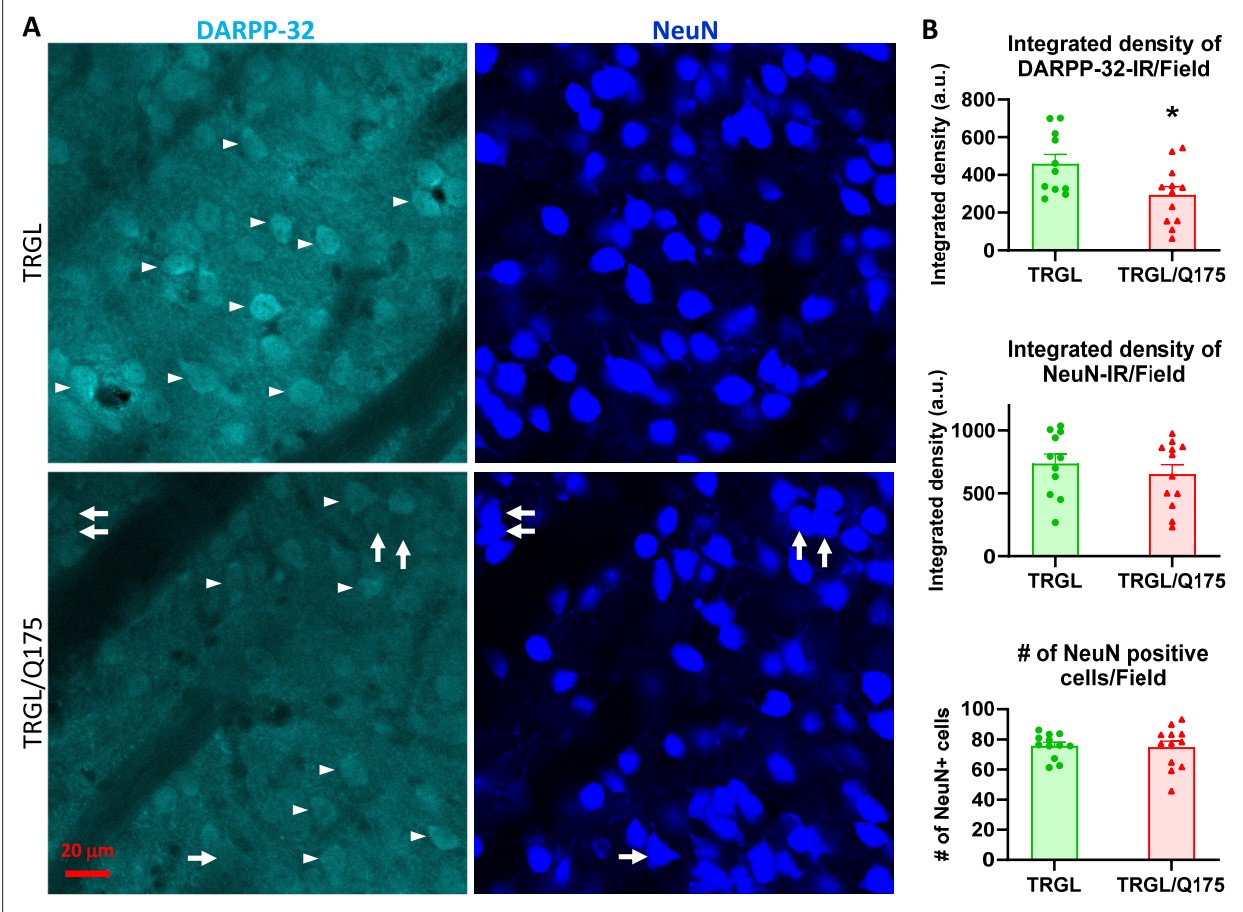

**Figure 4.** Decrease of DARPP-32-IR in striatal neurons of 17-mo-old TRGL/Q175 in the absence of a NeuN signal reduction. Brain sections from TRGL and TRGL/Q175 (n=10 mice/genotype, four sections/mouse) were double-immunostained with anti-DARPP-32 and -NeuN antibodies. Confocal images from the cranial-dorsal portion of the striatum (three images at 120 x/section) were collected and representative images are shown (**A**). Arrows depict examples of neurons showing strong NeuN signal while minimal DARPP-32-IR. (**B**) Images were quantified by Image J for Integrated Density of DARPP-32- (top) or NeuN-IR/Image field (middle), and for # of NeuN positive cells/Image field (bottom). Statistical significances between the two groups were analyzed by unpaired, two-tailed t-test. *p<0.05.

to Veh-treated TRGL/Q175 (*Figure 5A and B*). Additionally, there were also INK-induced changes in marker proteins located downstream of autophagy induction, including, for example, increased levels of p-ATG14 in INK-treated samples (*Figure 5A and B*). It is interesting that although INK did not induce any alterations in the transgene product tfLC3-I or -II, there was a trend of decreased endogenous LC3-I, leading to a statistically significant increase in the ratio of LC3-II/I, and such a result in LC3-I reduction was reproduced in a repeated experiment (*Figure 5A and B*). Thus, the pilot study in WT mice and the actual study in TRGL/Q175 mice together established mTORi INK's BBB permeability, target engagement, and ability to modify molecular events in the ALP.

## mTOR inhibition alters AV/LY subtypes

We took advantage of the tfLC3 construct reporting *in vivo* autophagy flux (*Lee et al., 2019*) to assess INK effects on AV/LY subtypes in TRGL/Q175. Confocal images from INK- or Veh-treated TRGL/Q175 and Veh-treated TRGL brain sections of 7-mo-old mice, immuno-stained with anti-CTSD antibody and detected with a third fluorophore, were collected (*Figure 6A*) and processed for hue-angle based quantitative analysis using the protocol described previously (*Lee et al., 2019*). The analysis in striatal neurons (*Figure 6B*) revealed reversal of the two main ALP alterations. First, the pre-existing abnormally lowered LY number in TRGL/Q175-Veh (*Figure 6B*, LY group, red vs green), was reversed by INK treatment (*Figure 6B*, LY group, red vs blue). Second, INK reduced the number of AL in TRGL/Q175-Veh compared to TRGL-Veh (*Figure 6B*, AL group, red vs blue), implying improved AL clearance

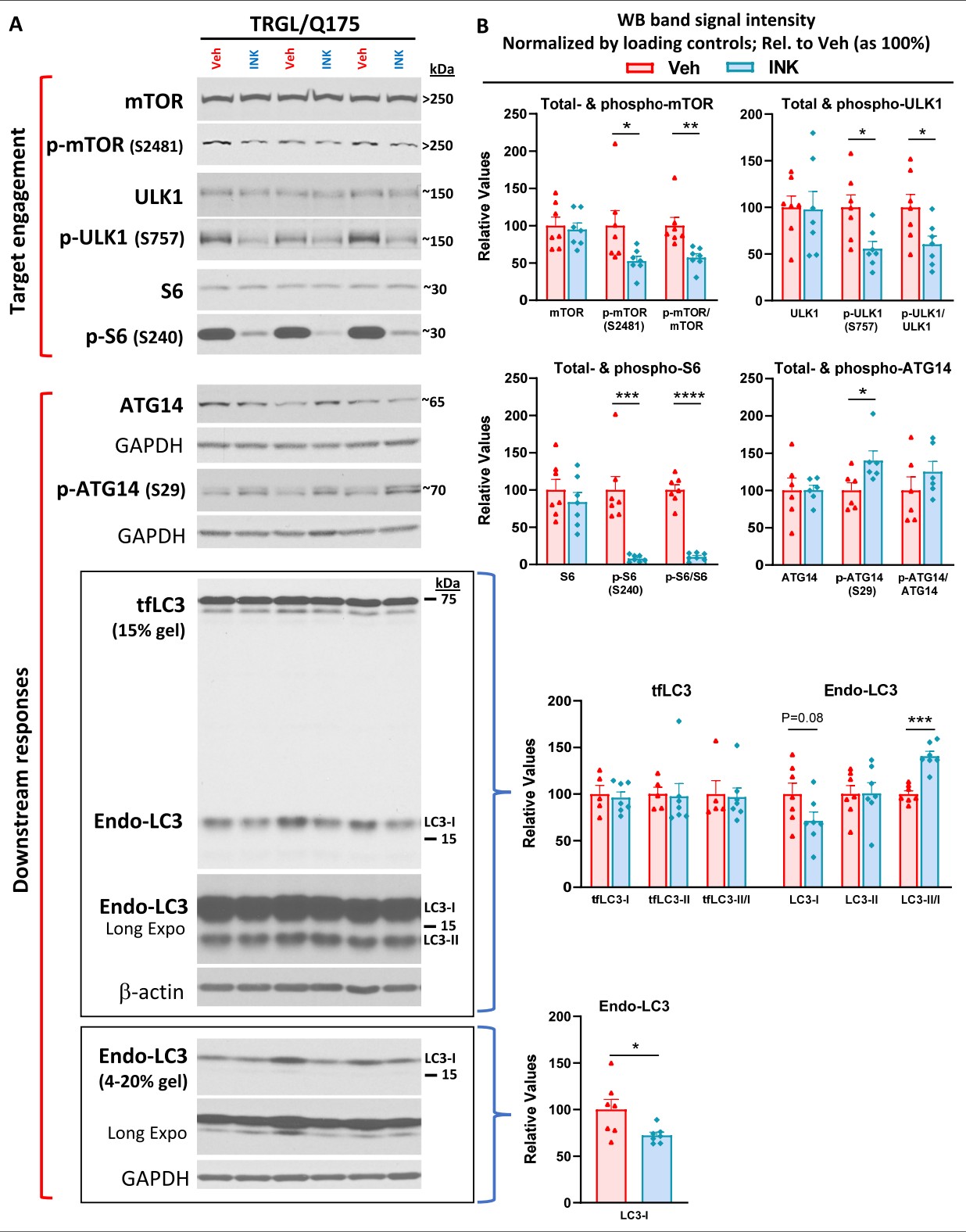

**Figure 5.** mTORi INK exhibits target engagement and induces downstream responses in the autophagic-lysosomal pathway (ALP) in 7-mo-old TRGL/Q175. Equal amounts of proteins from brain homogenates of 7-mo-old TRGL/Q175 mice untreated (labeled as 'Veh') or mechanistic target of rapamycin kinase inhibitor (mTORi) INK treated [4 mg/kg (4-mpk), daily, 3 wk; labeled as 'INK'] were subjected to SDS-PAGE and processed for western blotting (WB) with antibodies directed against several marker proteins in the autophagy pathway, representing target engagement of INK or downstream responses. Immunoblotting for each marker protein was performed one or more times depending on the quality of the blots. Representative blots

*Figure 5 continued on next page*

*Figure 5 continued*

are shown on the left (**A**) while quantitative results of the blots are shown on the right (**B**). The bottom LC3 blots represent a repeated immunoblotting experiment. Values are the Mean ± SEM for each group (n=6–7 mice per condition). Significant differences between the two groups were analyzed by unpaired, two-tailed t-test. *p<0.05, **p<0.01, ***p<0.001, ****p<0.0001. tfLC3=mRFP-eGFP-LC3; Endo-LC3=endogenous LC3.

The online version of this article includes the following source data and figure supplement(s) for figure 5:

**Source data 1.** Original western blots for *Figure 5A*.

**Source data 2.** Original western blots for *Figure 5A*, labeled for the relevant bands.

**Figure supplement 1.** Dosing tests demonstrate mechanistic target of rapamycin kinase inhibitor (mTORi) INK blood-brain barrier (BBB) penetration and target engagement even at low dosages.

**Figure supplement 1—source data 1.** Original western blots for *Figure 5—figure supplement 1*.

**Figure supplement 1—source data 2.** Original western blots for *Figure 5—figure supplement 1*, labeled for the relevant bands.

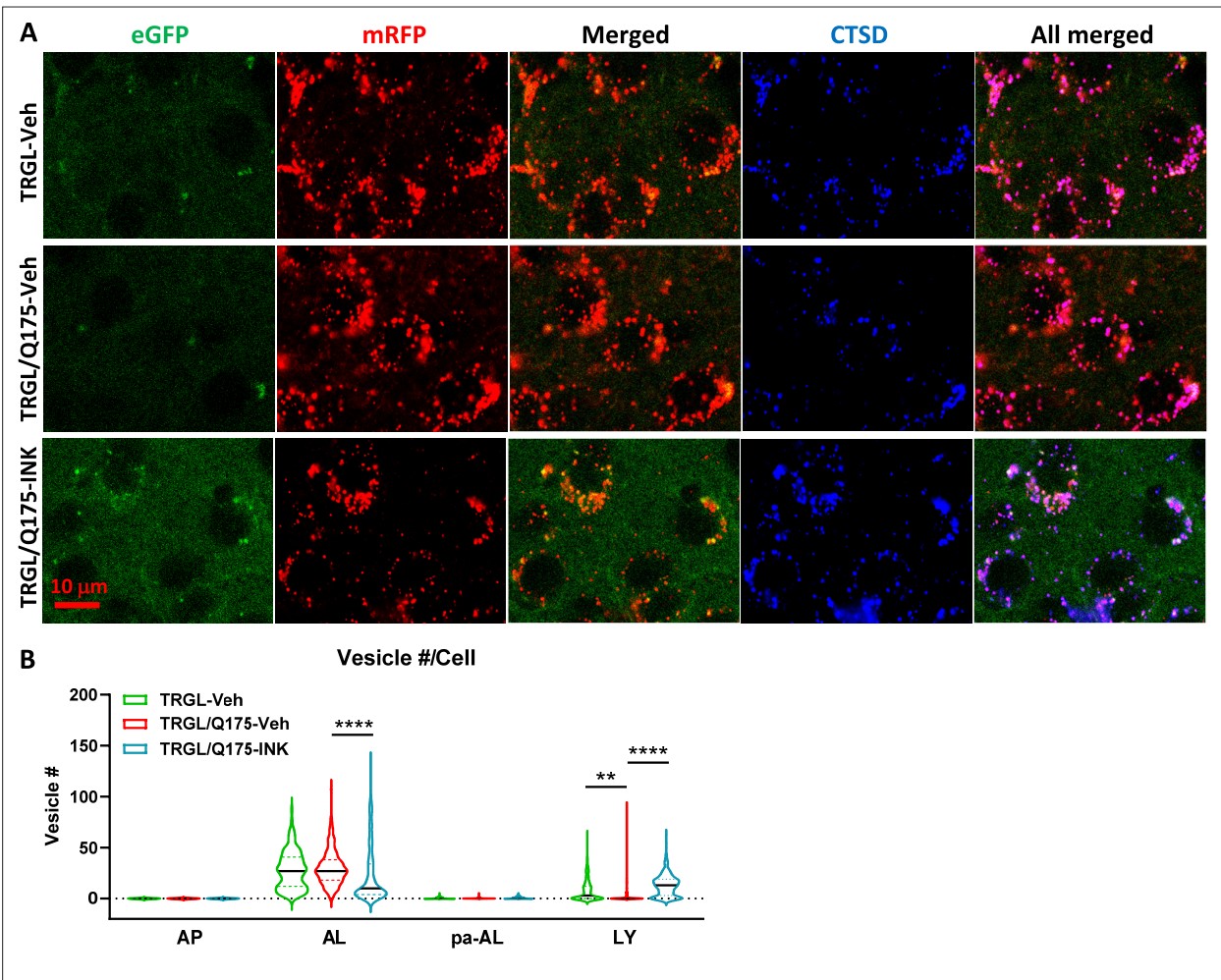

**Figure 6.** Mechanistic target of rapamycin kinase inhibitor (mTORi) INK reverses the mild alteration of autophagic vacuole (AV)/lysosomes (LY) subtypes in the striatum of 7-mo-old TRGL/Q175. (**A**) Brain sections from untreated or INK (4-mpk, 3 w)-treated TRGL/Q175 vs. TRGL (four sections/mouse, 5–6 mice per condition) were immunostained with an anti-cathepsin D (CTSD) antibody. Confocal images from the cranial-dorsal portion of the striatum (three images at 120 x/section) were collected and representative images for each eGFP-LC3 (green), mRFP-LC3 (red), and CTSD (blue) are shown. (**B**) Hue angle-based analysis was performed for AV/LY subtype determination using the methods described in *Lee et al., 2019*. Data are presented as Vesicle #/Neuron (TRGL-Veh: n=260 neurons; TRGL/Q175-Veh: n=218 neurons; TRGL/Q175-INK: n=287 neurons). Statistical significances among the groups were analyzed by one-way ANOVA followed by Sidak's multiple comparisons test. **p<0.01, ****p<0.0001.

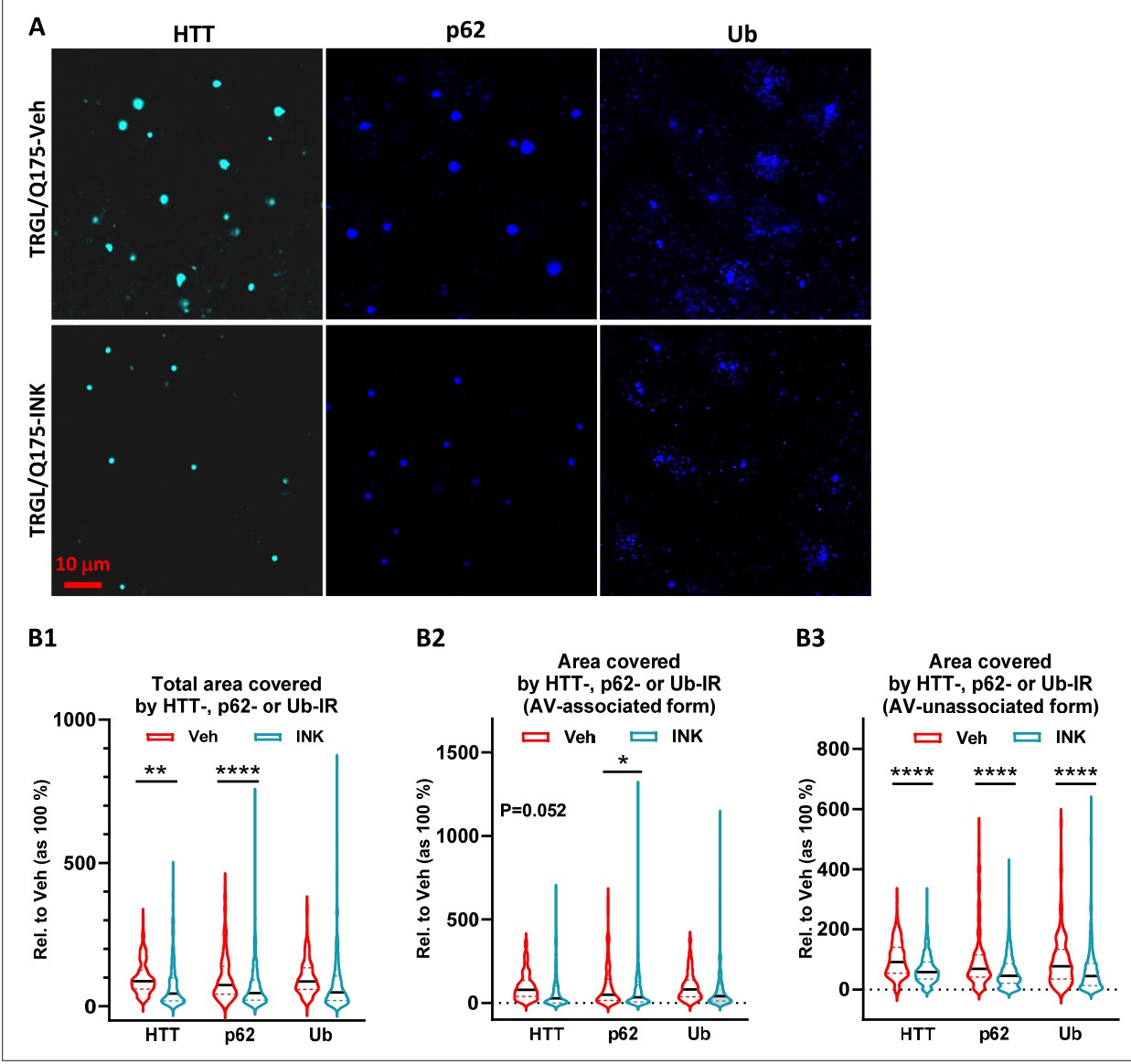

**Figure 7.** Mechanistic target of rapamycin kinase inhibitor (mTORi) INK reduces HTT-, Ub-, or p62-IR-covered areas parallelly in the striatum of 7-mo-old TRGL/Q175. (**A**) Brain sections from INK (4-mpk, 3 w)-treated or untreated TRGL/Q175 (four sections/mouse, 6–7 mice/condition) were immunostained with anti-HTT (MW8), -p62, or -pan-Ub antibodies. Confocal images from the cranial-dorsal portion of the striatum (three images at 120 x/section) were collected. Shown are single-channel images (i.e. without showing the eGFP and mRFP signals). (**B1–B3**) Areas covered by either the HTT-, p62-, or Ub-IR on a per cell basis are quantified (for HTT-IR, TRGL/Q175-Veh: n=196 neurons, TRGL/Q175-INK: n=385 neurons; for p62-IR, TRGL/Q175-Veh: n=311 neurons, TRGL/Q175-INK: n=378 neurons; for Ub-IR, TRGL/Q175-Veh: n=244 neurons, TRGL/Q175-INK: n=347 neurons) and grouped as 'Total Area' (**B1**), 'AV-associated Form' (i.e. the IR which was associated with tfLC3 signals representing autophagic vacuoles, AVs) (**B2**) or 'AV-unassociated Form' (i.e. the IR which was not associated with the tfLC3 signals) (**B3**). Statistical significances between the two groups were analyzed by unpaired t-test. Two-tailed p-value: *p<0.05, ****p<0.0001.

and likely providing the basis for restoration of normal LY number. Together, the data suggest that autophagy phenotype in 7-mo Q175 is measurable but milder than that in the 17-mo-old mice shown in *Figure 3A* and that INK has the potential to modulate the autophagy pathway and restore normal function at this mild disease stage. mTOR inhibition mitigates mHTT-related aggresome pathology.

Importantly, we found that HD-like pathology such as mHTT IBs and associated protein aggregation of adaptor proteins, p62, and Ub, was ameliorated by INK treatment. Confocal images demonstrated that mHTT positive particles/inclusions were substantially reduced in INK-treated 7-mo-old TRGL/Q175, compared to Veh-treated TRGL/Q175 mice (*Figure 7A*). Moreover, p62-IR or Ub-IR also

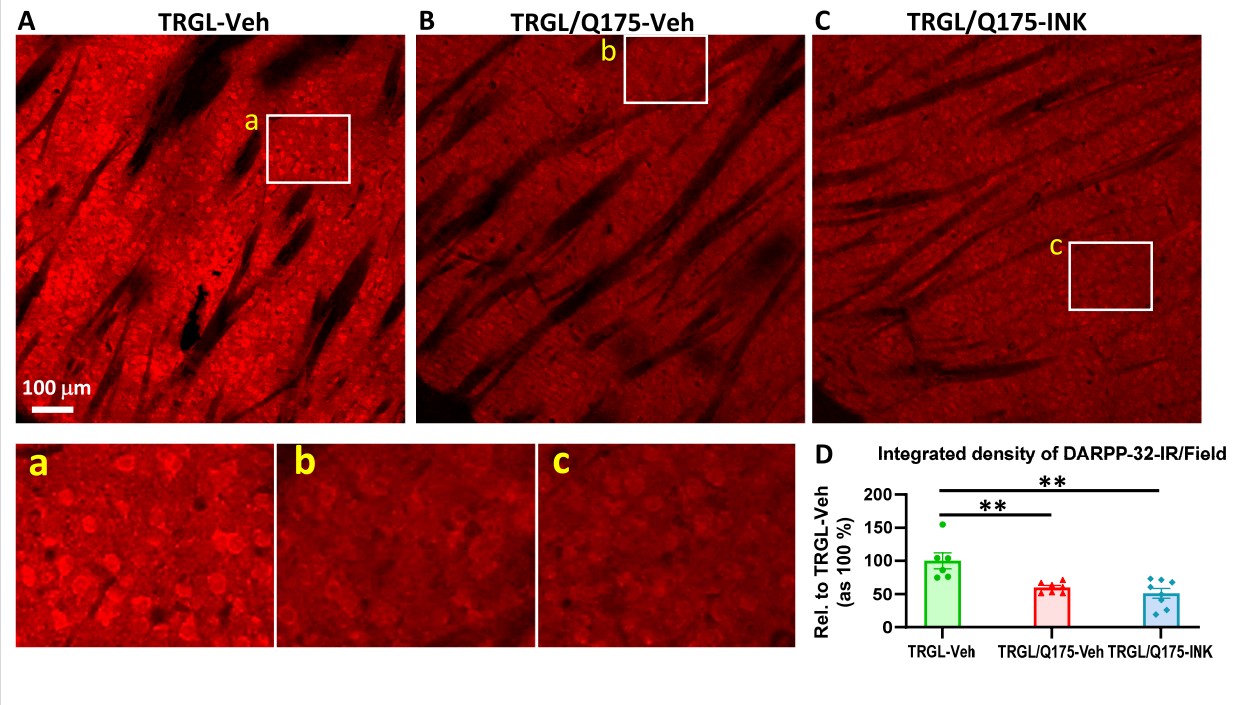

**Figure 8.** INK treatment does not reverse the reduction of DARPP-32-IR in the striatum of 7-mo-old TRGL/Q175. (**A-C**) Sagittal brain sections containing the striatum area were immunostained with an anti-DARPP-32 antibody, and 10 x images taken from the cranial-dorsal portion of the striatum are shown (first row). Boxed areas on the first row are enlarged and shown on the second row for easy viewing of the immunostaining patterns for each condition. (**D**) For quantitation purposes, each whole 10 x image (excluding the areas covered by the fiber bundles which usually exhibited minimal background staining – achieved by threshold setting) was quantified by ImageJ (one image/section, four sections/mouse) and the results are expressed as the Integrated Density of DARPP-32-IR. n=6–8 mice/condition. Statistical significances among the groups were analyzed by one-way ANOVA followed by Sidak's multiple comparisons test. **p<0.01 compared to TRGL-Veh.

changed in a similar trend (*Figure 7A*), consistent with the high degree of colocalization of mHTT/p62/Ub signals shown in *Figure 2A*. The corresponding quantitative analysis, shown as Area Covered by either mHTT-, p62-, or Ub-IR on a per cell basis, confirmed statistically significant decreases in the Total Area Covered by mHTT- and p62-signals (*Figure 7B1*). With the assistance of the endogenous tfLC3 signal (particularly the mRFP signal) in the TRGL to identify the association status of the mHTT-, p62-, or Ub-IR with AVs, the images were additionally quantified to generate the separated 'Area Covered by the AV-associated Form' (*Figure 7B2*) and 'Area Covered by the AV-unassociated Form' (*Figure 7B3*) of mHTT-, p62-, or Ub- signals. The results collectively suggest decreased mHTT-, p62-, or Ub-IR in treated compared to untreated TRGL/Q175, implying that INK treatment was able to promote the clearance of mHTT and related receptor proteins p62 and/or Ub.

## mTOR inhibition does not reverse preexisting DARPP-32-IR reduction in TRGL/Q175

Finally, we evaluated neurodegenerative changes by employing DARPP-32 IHC on brain sections of Veh-treated or INK-treated TRGL/Q175. Similar to what was found in 17-mo-old mice (*Figure 4*), the group of TRGL/Q175-Veh, even at this young age (7-mo), already exhibited a diminished DARPP-32-IR signal compared to that in TRGL-Veh (*Figure 8A and B*). However, no differences between the untreated and INK-treated TRGL/Q175 groups were established by quantitative analysis (*Figure 8A and B*). Thus, the result suggests a lack of effectiveness of INK in reversing this pre-existing phenotype, implying that earlier intervention may be necessary (*Marchionini et al., 2022*).

## Discussion

Previous studies in cell and animal models have documented roles of HTT and mHTT in autophagy in relation to HD [for reviews, see *Croce and Yamamoto, 2019*; *Klionsky et al., 2021*; *Martin et al., 2015*]. WT HTT participates in normal autophagy by releasing ULK1 via mTOR inhibition and serving as a scaffold to facilitate cargo sequestration by enhancing p62 interaction with LC3 and ubiquitinated cargos (*Ochaba et al., 2014*; *Rui et al., 2015*). mHTT in HD may activate autophagy by sequestering and thus inhibiting mTOR (*Ravikumar et al., 2004*). Most mHTT effects in cellular and mouse HD models appear to be inhibitory for autophagy induction steps, from initiation signaling and phagophore nucleation to cargo recognition/AP formation (*Ashkenazi et al., 2017*; *Martinez-Vicente et al., 2010*; *Pryor et al., 2014*; *Rui et al., 2015*; *Wold et al., 2016*). In addition, mHTT may disturb endo-lysosomal homeostasis by reducing exocytosis and promoting AL accumulation (*Zhou et al., 2021*). In the current study using Q175, we have found that mHTT is an ALP substrate and that the ALP is relatively intact, with some late-onset minor alterations, such as insufficient mTOR activity reflected by lowered phospho-p70S6K level and ratio of phospho-ULK1 (S757)/total ULK1, lysosome depletion, and AL/pa-AL elevation. This mild late-onset nature of ALP pathogenesis allowed us to modulate ALP with a mTOR inhibitor in 6-mo-old TRGL/Q175 which resulted in positive effects on rescuing Q175 phenotypes including normalization of AL/LY subtypes and reduction in aggresome pathology.

### ALP impairments in Q175 are mild and late-onset

In the current study, we assessed ALP alterations in the Q175 model by analyzing protein markers in all phases of the ALP including autophagy induction signaling, membrane nucleation/AP formation, autophagy adaptor proteins, and AL formation/substrate degradation. By immunoblot analysis, alterations in the tested marker proteins even at 17 mo of age in TRGL/Q175 mice were limited to a reduction in p-P70S6K (T389), similar to that reported previously (*Ravikumar et al., 2004*), and a reduced ratio of p-ULK1 (S757)/total ULK1 as a result of increased total ULK1. While these two alterations together would suggest a lower mTOR activity and autophagy activation, consequential alterations of downstream markers, particularly proteins in the ATG14-containing Beclin-1/VPS34 complex, were not detected. Although 4 out of the 10 tested TRGL/Q175 mice exhibited very low levels of p-ATG14, implying a deficit in this complex as previously reported (*Wold et al., 2016*), variation among animals was too great to draw statistical conclusions. In this regard, mice used in the current study were heterozygous, which results in less autophagy alteration than that reported in homozygous mice using certain autophagy markers (*Abd-Elrahman et al., 2017*). It may also be noted in our experience that immunoblot analysis of brain tissue averages contributions from varied cell types that may respond to disease states in opposite directions and mask changes in a given cell population. Thus, we also applied a neuron-specific analysis using our neuron-specific tfLC3 reporter and hue-angle-based image quantification approach, which yielded more sensitive detection and findings.

In brain sections from 17-mo-old TRGL/Q175 mice, we observed increased numbers of AL including the population of pa-AL and reduced numbers of LY. Such changes in AL/pa-AL were not readily observed in TRGL/Q175 mice at 2 mo (not shown) or 7 mo of age (e.g. *Figure 6B*, which only exhibited a reduction in LY number), indicating that even if these alterations signify an existence of impairments in the late phase of the ALP, they do not occur until a later age (e.g. 17 mo). Importantly, this pattern of late-onset ALP pathology progression, e.g., AL/pa-AL increases, is consistent with our findings from HD human brains (*Berg et al., 2024*) where enlarged CTSD-positive AL accumulate and cluster in affected neuronal populations at the late disease stages (HD3 and HD4). These findings suggest that the overall autophagy function in heterozygous Q175 is largely maintained until the relatively late disease stage and provide a rationale to stimulate autophagy as a therapeutic intervention when the autophagy machinery, especially those involved in clearance, is still functional, which would generate beneficial effects. We obtained support for this concept in the subsequent study with mTORi INK when administered to TRGL/Q175 at 6 mo of age for 3 wk.

### Autophagy as a primary mechanism for INK-enhanced clearance of small mHTT species

One of the key findings from this study is the reduction of mHTT-IR in neurons detected by immunolabeling after the 3 wk oral treatment with mTORi INK (*Figure 7*). This beneficial outcome from the drug treatment is interpreted as a primary result of the manipulation/stimulation of autophagy by the

compound, leading to enhanced autophagy clearance of mHTT species including smaller, presumably soluble, species and larger aggregates such as the IBs. This interpretation is supported by the following considerations and evidence. First, in theory, the compound used for the treatment is a mTORi, so it is expected to enhance autophagy (*Bensalem et al., 2021*; *Kim and Guan, 2015*). Second, target engagement is achieved as clearly reflected by the diminished levels of mTOR-mediated phosphorylation on mTOR itself (S2481), ULK1 (S757), and p-S6 (S240/S244), along with changes in proteins downstream of mTOR, such as p-ATG14 (S29) and LC3. Third, hue-angle based confocal image analysis of AL/LY subtypes reveals a reduced AL number and an increased LY number in INK-treated TRGL/Q175 compared to Veh-treated TRGL/Q175. The findings from this analysis of AL/LY subtypes highlight the advantages of crossing TRGL with Q175 mice to evaluate autophagy modulators on the ALP, consistent with a notion of developing better tools capable of investigating autophagy flux *in vivo* (*Nixon, 2013*; *Vodicka et al., 2014*).

Our confocal microscopy analyses revealed mHTT signal in CTSD-positive vesicles, which can be classified as AL based on their LC3 fluorescence, and IEM detected mHTT-silver-enhanced gold particles in AP and AL including lipofuscin granules. Both findings indicate the existence of mHTT species inside the ALP as autophagy substrates. The presence of low levels of mHTT in CTSD-positive AL also implies a constitutive functional degradation event of mHTT in AL (*Figure 2B1*, 10-mo-old mouse). This is in contrast to the case in young AD mouse models where early-onset autophagy-stress exhibited severe substrate accumulation (e.g. reflected by strong LC3 signal) due to deficits of degradative functions within AL. (*Lee et al., 2022*). A reduced level in the portion of AV-associated mHTT signals (*Figure 7B2*) by mTORi INK further suggests that mHTT degradation has been enhanced after manipulating the ALP. This AV-associated mHTT pool is considered to be corresponding to the mHTT-silver-enhanced gold signal under IEM (*Figure 2B2*), i.e., amorphous and existing within the lumen of AV. Such mHTT localization and ultrastructural features are consistent with recent findings that mHTT proteins exogenously added to Q175 brain sections for incubation are recruited to multivesicular bodies/amphisomes, AL, and residual bodies (lipofuscin granules), and ultrastructurally localized to non-fibrillar, electron-lucent regions within the lumen of these organelles (*Zhou et al., 2021*).

## Possible autophagy-related mechanism for the clearance of large mHTT IBs

It is notable that the AV-non-associated form of mHTT (i.e. did not show colocalization with the tfLC3 signal implying that they might not have contact with AVs) decrease more obviously after drug treatment, along with the reductions in p62 and Ub signals (*Figure 7B3*). These mHTT species are mainly larger aggresomes/IBs, including those within the nuclei. Based on our mHTT-IEM observation (*Figure 1B*), the IBs in Q175 brain are fibrillar aggregates in the nucleus and cytoplasm, exhibiting a homogeneous feature in shape (i.e. round/oval cotton-ball) and in content (i.e. short fine fibrils), similar to the nuclear IBs we found in human HD brains (*Berg et al., 2024*), whose cytoplasmic and neuritic IBs, however, also exhibit more heterogeneous compositions such as a mixture of fine fibrils with AVs, fingerprint-like structures or bundles of microtubules. In the current study, we did not observe that a whole IB is contained within an AP or AL in our EM examination, raising a question of how these larger aggregates could be degraded by the autophagy machinery after an autophagy activation by mTOR inhibition. This issue is informed by findings that the mHTT species are dynamic and the aggregates can form as different phases (liquid-like or solid-like), making it possible for mHTT species to move out or into IBs (*Aktar et al., 2019*; *Liu et al., 2015*; *Peskett et al., 2018*; *Riguet et al., 2021*) and to shuttle between the nucleus and the cytoplasm (*Atwal et al., 2007*; *Xia et al., 2003*). Thus, it is speculated that under the autophagy-activated condition induced by mTOR inhibition, the enhanced clearance of mHTT in the ALP pool would promote movements of mHTT species from the IB forms, leading to a reduction on the overall mHTT load.

## Possible additional mechanisms for the reduction of mHTT species

Even if INK is a mTORi mainly targeting autophagy, we cannot completely exclude additional contributions from the UPS for the observed reduction of mHTT species given the crosstalk between the two proteolytic systems. It is well accepted that autophagy plays a more important role in the clearance of larger protein aggregates such as aggresomes/IBs (*Sarkar and Rubinsztein, 2008a*). For example, autophagy is involved in the degradation of p62 bodies (*Bjørkøy et al., 2005*) and autophagy markers

are colocalized with cytoplasmic mHTT IBs (*Iwata et al., 2005*). However, there is also evidence for degradation of polyQ aggregates in the nuclei (*Iwata et al., 2009*), which contain p62 condensates as a hub for UPS-mediated nuclear protein turnover (*Fu et al., 2021*). We observed a high level of colocalization of mHTT with p62 and Ub in aggregates at both intranuclear and extranuclear locations (*Figure 2A*), which is presumably more related to the UPS hub and autophagy, respectively. It is speculated that the reduction of mHTT in the cytoplasm due to the stimulation of autophagy may partially relieve the burden of the proteasome in both the cytoplasm and the nucleus so that the nuclear proteasome operates more effectively. One of the observations reported here which may support the above speculation is the reductions of AV-non-associated forms of mHTT/p62/Ub (*Figure 7B3*), given that some of these aggregates should exist within the nucleus and, therefore, their reduced levels (in the nucleus) may reflect increased intranuclear UPS activity, besides the other possibility that they may travel from the nucleus to the cytosol for clearance (by autophagy) as discussed above.

Lowering mHTT via interfering protein production (e.g. through RNAi, antisense oligonucleotides) has been an attractive strategy in HD therapeutic development (*Kordasiewicz et al., 2012*; *Tabrizi et al., 2019*). Given that mTOR regulates multiple cellular pathways including protein synthesis (*Magnuson et al., 2012*; *Wang and Proud, 2006*), the inhibition of mTOR as was done in the present study would potentially affect protein synthesis as well. Thus, while our results of decreases in mHTT signals (*Figure 7*) can be interpreted as a result of autophagy-mediated clearance of mHTT, a possibility cannot be excluded that mTOR inhibition may result in a reduction in HTT production which may also contribute to the observed results – future studies should determine how significant such a contribution is.

In summary, Q175 models, as detected in the current study, develop ALP alterations late in the disease progression. Although milder than what is seen in late-stage HD human brain (*Berg et al., 2024*), there are similarities in ALP pathobiology between human HD brains and Q175 brains. These include the late-onset nature, mHTT as an ALP substrate, increased AL/pa-AL, mouse IB ultrastructural feature similar to the common fine fibril type of human IBs. The late-onset mild ALP pathology in Q175 is different from the severe AV accumulation in mouse models of AD as mentioned above (*Lee et al., 2022*). Together, the mild and late-onset nature of ALP alterations in Q175 provides a basis for manipulating autophagy to be a promising therapeutic strategy if the manipulations (e.g. stimulating autophagy with mTORi as in this study) are applied when the autophagy machinery is largely undamaged. This has been validated to be successful in this study as demonstrated by the INK-induced reduction of mHTT species along with other beneficial effects, consistent with findings from similar studies and supporting the general notion of lowering mHTT as a therapeutic strategy for HD (*Barker et al., 2020*; *Cortes and La Spada, 2014*; *Ravikumar et al., 2004*). However, it should be noted that the current study is an experimental therapeutic attempt in a mouse model which, consistent with previous reports (*Ravikumar et al., 2004*), is just a proof of concept for manipulating autophagy (e.g. via inhibiting mTOR in the current setting) as a potential therapeutic. The clinical implications from such studies require further rigorous verifications where early diagnosis and earlier interventions would be critical factors to be considered.

## Materials and methods

**Key resources table**

| Reagent type (species) or resource | Designation | Source or reference | Identifiers | Additional information |
|---|---|---|---|---|
| Genetic reagent, KI-mouse (*Mus musculus* male and female) | Q175 KI | CHDI-81003003 | N/A | Knock-in Mouse human *HTT* exon 1 sequence containing a ~190 CAG repeat tract |
| Genetic reagent, Transgenic (*Mus musculus* male and female) | TRGL6 | *Lee et al., 2019* | N/A | Transgenic with Thy1 promoter |
| Genetic reagent, Transgenic (*Mus musculus* male and female) cross with KI (*Mus musculus* male and female) | Q175/TRGL | This study | N/A | Crossed the zQ175 KI with the TRGL6 mice |

*Continued on next page*

*Continued*

| Reagent type (species) or resource | Designation | Source or reference | Identifiers | Additional information |
|---|---|---|---|---|
| Sequence-based reagent | Forward primer for genotyping TRGL: 50-CTT TCC CCA CAG AAT CCA AGT CGG AAC-30 | *Lee et al., 2019* | | |
| Sequence-based reagent | Reverse primer for genotyping TRGL: 50-GCA CGA ATT CGG GCG CCG GTG GAG TGG CGG-30 | *Lee et al., 2019* | | |
| Antibody | tHTT | Cell Signaling Technology | Cat# 5656; RRID:AB_10827977 | WB: 1:1000 |
| Antibody | ntHTT mEM48 | Millipore sigma | Cat# MAB5374; RRID:AB_177645 | WB 1:200 IHC 1:50 |
| Antibody | HTT 1C2 | Millipore sigma | Cat# mab1574; RRID:AB_94263 | WB 1:1000 |
| Antibody | HTT human | Millipore sigma | Cat# mab5490; RRID:AB_2233522 | WB 1:1000 IHC 1:200 |
| Antibody | HTT MW8 | Develop Studies Hybridoma Bank, University of Iowa | Cat# MW8; RRID:AB_528297 | |
| Antibody | HTT PHP2 | CHDI/Coriell | Cat# CHDI-90001516–2 | |
| Antibody | MTOR | Cell Signaling Technology | Cat# 2983; RRID:AB_2105622 | WB: 1:1000 |
| Antibody | p-MTOR S2481 | Cell Signaling Technology | Cat# 2874 | WB 1:1000 |
| Antibody | p-MTOR S2448 | Cell Signaling Technology | Cat# 2971; RRID:AB_330970 | WB: 1:1000 |
| Antibody | p70S6K | Cell Signaling Technology | Cat# 2708; RRID:AB_390722 | WB: 1:1000 |
| Antibody | p-p70S6K S371 | Cell Signaling Technology | Cat#: 9208 | WB: 1:1000 |
| Antibody | p-p70S6K T389 | Cell Signaling Technology | Cat# 9234; RRID:AB_2269803 | WB: 1:1000 |
| Antibody | S6 ribosomal | Cell Signaling Technology | Cat# 2317; RRID:AB_2238583 | WB: 1:1000 |
| Antibody | p-S6 ribosomal S240/S244 | Cell Signaling Technology | Cat# 5364; RRID:AB_10694233 | WB: 1:1000 |
| Antibody | ULK1 | Cell Signaling Technology | Cat# 6439; RRID:AB_11178933 | WB: 1:500 |
| Antibody | p-ULK S757 | Cell Signaling Technology | Cat# 6888; RRID:AB_10829226 | WB: 1:500 |
| Antibody | p-ULK S317 | Cell Signaling Technology | Cat# 37762; RRID:AB_2922992 | WB: 1:500 |
| Antibody | ATG5 | Cell Signaling Technology | Cat# 12994; RRID:AB_2630393 | WB: 1:1000 |
| Antibody | ATG5 | Millipore sigma | Cat# abc14 | WB: 1:750 IHC 1:500 |
| Antibody | ATG14 | Cell Signaling Technology | Cat# 5504; RRID:AB_10695397 | WB: 1:1000 |
| Antibody | p-ATG14 S29 | Cell Signaling Technology | Cat# 92340; RRID:AB_2800182 | WB: 1:1000 |

*Continued*

| Reagent type (species) or resource | Designation | Source or reference | Identifiers | Additional information |
|---|---|---|---|---|
| Antibody | Beclin 1 | BD Bioscience | Cat# 612113; RRID:AB_399484 | WB: 1:1000 |
| Antibody | p-Beclin 1 S30 | Cell Signaling Technology | Cat# 35955 | WB: 1:1000 |
| Antibody | VPS34 | Cell Signaling Technology | Cat# 4263 | WB 1:1000 |
| Antibody | p-VPS34 S249 | Cell Signaling Technology | Cat# 13857; RRID:AB_2798332 | WB 1:1000 |
| Antibody | TRAF6 | Cell Signaling Technology | Cat# 8028 | WB 1:1000 |
| Antibody | LC3 | Millipore sigma | Cat# abc929 | WB 1:500 IHC 1:100 |
| Antibody | LC3 | Novus Biologics | Cat# NB100-2220; RRID:AB_10003146 | WB 1:1000 IHC 1:200 |
| Antibody | p62/sqstm1 | BD Biosciences | Cat# 610832; RRID:AB_398152 | WB 1:2000 IHC 1:200 |
| Antibody | p62/SQSTM1 c-term | Progen Biotechnik | Cat# GP62-C; RRID:AB_2687531 | WB 1:1000 IHC 1:250 |
| Antibody | Ubiquitin | Dako Agilent | Cat# z0458; RRID:AB_2315524 | |
| Antibody | Ubiquitin | Abcam | Cat# ab7780; RRID:AB_306069 | |
| Antibody | LAMP1 | Develop Studies Hybridoma Bank, University of Iowa | Cat# H4A3; RRID:AB_2296838 | IHC 1:50 |
| Antibody | CTSB | Neuromics | Cat# GT15047; RRID:AB_2737184 | WB 1:5000 IHC 1:500 |
| Antibody | CTSD | In house | RU4 | WB 1:10,000 IHC 1:5000 |
| Antibody | CTSD sheep | In house | D-2–3 | WB 1:1000 IHC 1:500 |
| Antibody | DARPP32 | Abcam | Cat# ab40801; RRID:AB_731843 | WB 1:1000 IHC 1:100 |
| Antibody | NeuN | Millipore Sigma | Cat# mab377; RRID:AB_2298772 | IHC 1:400 |
| Antibody | β–actin | Millipore Sigma | Cat# A1978; RRID:AB_476692 | WB 1:10,000 |
| Antibody | Goat anti-rabbit secondary | Vector Laboratories | Cat# 68–4140 | IHC: 1:500 |
| Antibody | Goat anti-mouse secondary | Vector Laboratories | Cat# BA-9200 | IHC: 1:500 |
| Antibody | Alexa Fluor 488-conjugated goat anti-rabbit IgG | Thermo Fisher Scientific | Cat# A11034; RRID:AB_2576217 | IHC: 1:500 |
| Antibody | Alexa Fluor 568- goat anti-rabbit IgG | Thermo Fisher Scientific | Cat# A11036; RRID:AB_10563566 | IHC: 1:500 |
| Antibody | Alexa Fluor 647- goat anti-rabbit IgG | Thermo Fisher Scientific | Cat# A21245; RRID:AB_2535813 | IHC: 1:500 |
| Antibody | Alexa Fluor 405- goat anti-rabbit IgG | Thermo Fisher Scientific | Cat# A48254; RRID:AB_2890548 | IHC: 1:500 |
| Antibody | Alexa Fluor 568- goat anti-mouse IgG | Thermo Fisher Scientific | Cat# A11031; RRID:AB_144696 | IHC: 1:500 |
| Antibody | Alexa Fluor 647- goat anti-mouse IgG | Thermo Fisher Scientific | Cat# A21235; RRID:AB_2535804 | IHC: 1:500 |

*Continued*

| Reagent type (species) or resource | Designation | Source or reference | Identifiers | Additional information |
|---|---|---|---|---|
| Antibody | Alexa Fluor 568- goat anti-rat IgG | Thermo Fisher Scientific | Cat# A21247; RRID:AB_141778 | IHC: 1:500 |
| Antibody | Donkey anti-Rabbit IgG HRP | Jackson ImmunoResearch | Cat# 711-035-152; RRID:AB_10015282 | WB: 1:5000 |
| Antibody | Donkey anti-Mouse IgG HRP | Jackson ImmunoResearch | Cat# 712-035-150; RRID:AB_2340638 | WB: 1:5000 |
| Antibody | Donkey anti-goat IgG HRP | Jackson ImmunoResearch | Cat# 705-035-003; RRID:AB_2340390 | WB: 1:5000 |
| Antibody | 10 nm gold anti mouse IgG | Electron Microscopy Sciences | Cat# 25129 | IEM 1:50 |
| Antibody | 10 nm gold anti rabbit IgG | Electron Microscopy Sciences | Cat# 25109 | IEM 1:50 |
| Chemical compound, drug | INK (mTOR i) | ChemScene | CAS 1224844-38-5 | Dissolved in 0.5% carboxymethyl cellulose (Sigma, Cat #5678) and 0.05% tween 80 in water |
| Chemical compound, drug | PBS | Thermo Fisher Scientific | Cat# BP339-4 | |
| Chemical compound, drug | Sodium cacodylate buffer | Electron Microscopy Sciences | Cat#11652 | 0.1 M for fixation of brains |
| Chemical compound, drug | Paraformaldehyde | Electron Microscopy Sciences | Cat#15714 | 4% for fixation of brains |
| Chemical compound, drug | 25% glutaraldehyde | Electron Microscopy Sciences | Cat# 16220 | |
| Chemical compound, drug | Uranyl acetate | Electron Microscopy Sciences | Cat# 22400–4 | EM processing |
| Chemical compound, drug | Lead citrate | Electron Microscopy Sciences | Cat#22410 | EM processing |
| Chemical compound, drug | Osmium tetroxide | Ted Pella | Cat#18465 | EM processing |
| Chemical compound, drug | Sodium metaperiodate | Sigma-Aldrich | Cat#S1878-25g | EM processing |
| Chemical compound, drug | Permount | Electron Microscopy Sciences | Cat# 17986 | Dab IHC |
| Chemical compound, drug | Flouro-gel | Electron Microscopy Sciences | Cat# 17985–10 | IF IHC |
| Chemical compound, drug | FBS | Thermo Fisher Scientific | Cat# 26140 | IHC blocking |
| Chemical compound, drug | Horse serum | Thermo Fisher Scientific | Cat# 16050 | IHC blocking |
| Chemical compound, drug | Spurr resin | Electron Microscopy Sciences | Cat# 14300 | |
| Commercial assay kit | Vectastain ABC | Vector Laboratories | Cat# PK-4000; RRID:AB_2336818 | For DAB staining |
| Commercial assay kit | DAB Peroxidase Substrate kit | Vector Laboratories | Cat# SK-4100; RRID:AB_2336382 | For DAB staining |
| Commercial assay kit | HQ silver kit | NanoProbes | Cat# 2012–45 ml | EM processing |
| Software | ImageJ | NIH | https://imagej.nih.gov/ij/ | |
| Software | Excel | Microsoft | Microsoft 365 | |
| Software | GraphPad Prism 8.0.1 | GraphPad | | |
| Other | Nitrocellulose membrane | Whatman | | WB 0.2µm-pore |
| Other | Aclar embedding film | Electron Microscopy Sciences | Cat# 50425–25 | EM processing |
| Other | 75 mesh nickel grids with carbon coating and formvar | Electron Microscopy Sciences | Cat# pi-75-ni-25 | EM processing |
| Other | EM block | Electron Microscopy Sciences | Cat# 25596 | EM processing |

## The HD Q175 mouse model and the generation of a new Q175 model crossed with the autophagy reporter mouse TRGL (TRGL/Q175)

Q175 mice (B6J.zQ175 Knock-In mice, CHDI-81003003) (*Menalled et al., 2012*; *Menalled et al., 2003*) were obtained from the Jackson Lab (Stock No. 027410) on a C57BL/6 J background. The zQ175 KI allele has the mouse *Htt* exon 1 replaced by the human *HTT* exon 1 sequence containing a~190 CAG repeat tract, and the average size for the mice used in this study was 200 CAG. Q175 genotyping and CAG sizing were conducted by Laragen, Inc (Culver City, CA). The TRGL (**T**hy-1 m**R**FP-e**G**FP-**L**C3) mouse model, expressing tandem fluorescence-tagged LC3 (tfLC3 or mRFP-eGFP-LC3) in neurons, was generated, genotyped by PCR and maintained in a C57BL/6 J background as described previously (*Lee et al., 2019*). The TRGL was crossed with Q175 to generate the new TRGL/Q175 model featuring the tfLC3 probe as an *in vivo* autophagy reporter.

The mice were maintained in the Nathan Kline Institute for Psychiatric Research (NKI) animal facility and housed in a 12 hr light/dark cycle. All animal procedures were performed following the National Institutes of Health Guidelines for the Humane Treatment of Animals, with approval from the Institutional Animal Care and Use Committee at the NKI (AP2018-624). Animals of both sexes were used in this study. Details for mouse ages are in the Figure Legend and/or Results. All efforts were made to minimize animal suffering and the number of animals used. Mouse sample sizes were estimated based on similar experimental procedures in our prior studies (*Lee et al., 2019*; *Lee et al., 2022*; *Yang et al., 2017*; *Yang et al., 2011*; *Yang et al., 2014*), where a minimum of 5 mice per each mouse genotype per experimental condition would yield statistically significant differences in revealing alterations of autophagy in AD transgenic animal models vs wild-type mice.

## mTORi INK administration to mice

INK-128 (ChemScene) was formulated in the sterile filtered vehicle (Veh) – 0.5% carboxymethyl cellulose (Sigma, Cat #5678) and 0.05% Tween 80 in water – at different concentrations to achieve the various desired dosages. The mixture was homogenized with a tissue homogenizer, stored in a sterile container at room temperature for up to 2 wk, and stirred prior to each dosing. It was administered to mice via oral gavage at a dose volume of 5 ml/kg, which was a relatively smaller volume of drug solution (generally 100–150 µl depending on the mouse body weight) aimed at avoiding the suppression of appetite. Mice receiving the Veh (also at a solution volume of 5 ml/kg) served as the control group. Randomization of grouping: mice from each pool of sex and genotype of a particular age were pulled at random and put into either vehicle or drug treatment groups to make sure of even distribution of sexes and genotypes in the vehicle and drug treatments. Oral gavage was conducted with 20 G-38mm Cadence Science Malleable Stainless-Steel Animal Feeding Needles #9921 (Fisher Sci Cat #.14-825-275). Each of the INK-128 dosing tests (ranging from 2.5 to 10 mg/kg [mpk] with various durations) and the 3 wk daily 4 mpk treatment procedures were performed once.

## mTORi INK pharmacokinetics in mouse brains after oral administration

INK was formulated into 0.5% carboxymethyl cellulose and 0.05% Tween-80 in water to be administered orally. 6 mo old C57B6/J male mice were treated with compound at 1, 3, or 10 mpk or vehicle. 6, 12, or 24 hr after administration, mice were terminally anesthetized with pentobarbital, and cerebellum was dissected and flash frozen on liquid nitrogen and stored at –80 °C until analysis. Tissue was homogenized and extracted with acetonitrile (containing 0.1% formic acid). Extract was analyzed using a LC-MS/MS (Waters Xevo TQ MS) method with a lower limit of quantitation of 13 nM.

## Brain tissue preparation

To obtain tissues for experiments, the animals were anesthetized with a mixture of ketamine (100 mg/kg BW) and xylazine (10 mg/kg BW). Mice for light microscopic (LM) analyses were usually fixed by cardiac perfusion using 4% paraformaldehyde (PFA) in 0.1 M sodium cacodylate buffer [pH 7.4, Electron Microscopy Sciences (EMS), Hatfield, PA]. Following perfusion fixation, the brains were immersion-fixed in the same fixative overnight at 4 °C. For transmission electron microscopic (EM) study, 4% PFA was supplemented with 2% glutaraldehyde (EMS). For biochemical analyses such as immunoblotting, the brains were flash frozen on dry ice and stored at –70 °C. When both morphological and biochemical analyses were to be performed on the same brain, the brain was removed after

brief perfusion with saline. One hemisphere was frozen at –70 °C and the other half was immersion-fixed in 4% PFA for 3 d at 4 °C.

## Antibodies for immunohistochemistry (IHC) and western blotting (WB)

The following primary antibodies were used in this study. (1) from Cell Signaling Technology: tHTT rabbit mAb (total HTTs, #5656), MTOR rabbit mAb (#2983); p-MTOR (S2481 autophosphorylation) pAb (#2974), p-MTOR (S2448) pAb (#2971), p70S6K rabbit mAb (#2708), p-p70S6K (S371) pAb (#9208), p-p70S6K (T389) rabbit mAb (#9235), S6 ribosomal protein mAb (#2317), p-S6 ribosomal protein (S240/244) rabbit mAb (#5364), ULK1 pAb (#4773), ULK1 rabbit mAb (#6439), p-ULK1 (S757) pAb (#6888), p-ULK1 (S317) pAb (#37762), ATG5 rabbit mAb (#12994), ATG14 pAb (#5504), p-ATG14 (S29) rabbit mAb (#92340), p-Beclin 1 (S30) rabbit mAb (#35955), VPS34 rabbit mAb (#4263), p-VPS34 (S249) pAb (#13857), TRAF6 rabbit mAb (#8028). (2) from Millipore-Sigma: ntHTT mAb (mEM48, #MAB5374; N-Terminus-specific), HTT mAb (1C2, #MAB1574; epitope: N-terminal part of the human TATA Box Binding Protein (TBP) containing a 38-glns stretch), HTT mAb (#MAB5490; epitope: human HTT aa115-129), ATG5 pAb (#ABC14), LC3 pAb (#ABC929), NeuN (#MAB377), β–actin mAb (#A1978). (3) from other vendors: HTT mAb MW8 (epitope: HD exon 1 with 67Q; Develop Studies Hybridoma Bank, University of Iowa), HTT mAb PHP2 (CHDI-90001516–2, Coriell/CHDI), Beclin 1 mAb (BD Biosciences, #612113), LC3 pAb (Novus Biologics, #NB100-2220), p62/SQSTM1 mAb (BD Biosciences, #610832) or C-term-specific p62/SQSTM1 Guinea Pig pAb (Progen Biotechnik #C-1620); ubiquitin pAb (Dako Agilent, #Z0458), ubiquitin pAb (Abcam, #ab7780), DARPP32 (Abcam, #40801), LAMP1 rat mAb (Developmental Studies Hybridoma Bank, #H4A3), CTSB goat pAb (Neuromics, #GT15047). (4) In-house made: CTSD pAb (RU4). CTSD sheep pAb (D-2–3) (*Cataldo et al., 1990*).

The following secondary antibodies and reagents for immunoperoxidase labeling were purchased from Vector Laboratories (Burlingame, CA): biotinylated goat anti-rabbit or -mouse IgG/IgM, Vectastain ABC kit (PK-4000), and DAB Peroxidase Substrate Kit (SK-4100). Mouse on Mouse (M.O.M) detection kit (BMK-2201), normal-goat (S-100), and normal-donkey (S-2000–20) serum blocking solution were also from Vector Lab. The following secondary antibodies for immunofluorescence were purchased from Thermo Fisher Scientific (Waltham, MA): Alexa Fluor 488-conjugated goat anti-rabbit IgG (A11034), Alexa Fluor 568- goat anti-rabbit IgG (A11036), Alexa Fluor 647- goat anti-rabbit (A21245), Alexa Fluor Plus 405- goat anti-rabbit (A48254), Alexa Fluor 568- goat anti-mouse IgG (A11031), Alexa Fluor 647- goat anti-mouse (A21235), and Alexa Fluor 647- goat anti-rat (A21247). HRP-linked secondary antibodies for immunoblotting were obtained from Jackson ImmunoResearch Laboratories (West Grove, PA): Rabbit IgG (711-035-152), Mouse IgG (711-035-150), Rat IgG (712-035-150), and Goat IgG (705-035-003).

## Immunolabeling of brain sections

Immunoperoxidase and immunofluorescence IHC were performed according to the protocols previously described (*Lee et al., 2019*; *Yang et al., 2009*). Brain vibratome sections (40 µm) were blocked and incubated in primary antibody O/N (up to 3 d in some cases) at 4 °C. ABC detection method was used for immunoperoxidase labeling visualized with DAB while Alexa-Fluor conjugated secondary antibodies were used for immunofluorescence. Autofluorescence was quenched with 1% Sudan black (Sigma-Aldrich; St. Louis, MO) in 70% ethanol for 20 min. DAB-labeled sections was inspected on a Zeiss AxioSkop II equipped with a HrM digital camera (Carl Zeiss, Germany). Immunofluorescently labeled sections were examined on a Zeiss LSM880 confocal microscope. Independent investigator(s) coded animals to blind investigators when imaging and quantifying.

## Confocal image collection and hue-angle based quantitative analysis for AV/LY subtypes

Confocal imaging was performed using a plan-Apochromat 20 x or 40 x/1.4 oil objective lens on a LSM880 laser scanning confocal microscope with the following parameters: eGFP (ex: 488, em: 490–560 with MBS 488), mRFP (ex: 561, em: 582–640 with MBS 458/561), Alexafluor 647 (ex: 633, em: 640–710 with MBS 488/561/633) and DAPI (ex: 405, em: 410–483), with the 'best signal scanning mode' which separates scanning tracks for each excitation and emission set to exclude crosstalk between each fluorophore signal. The resolution of 40 x images was 1024×1024 pixels (corresponding to an area of 212.34×212.34 µm$^2$), and the resolution of 3 x digitally zoomed images was also 1024×1024

pixels (corresponding to an area of 70.78×70.78 $\mu m^2$). Detailed settings for image collection were reported previously (*Lee et al., 2019*).

Hue angle-based vesicle analysis enables quantitative determination of AV/LY subtypes including AP, AL, poorly acidified AL (pa-AL), and pure LY and the method was described in detail previously (*Lee et al., 2019*). Briefly, high resolution confocal images containing the LC3 (red and green) and CTSD (blue) punctate signals were analyzed with the Zen Blue Image Analysis Module from Carl Zeiss Microscopy. Threshold for each of the three-color channels (red, green, blue; RGB) was set by taking the average of intensity value from 20 neuronal perikarya. The signal was segmented into discrete puncta by using the automatic watershed function to separate clumped vesicles into individual puncta. Background signal was eliminated using the size exclusion function of Zen Blue. The R, G, and B intensity values of each vesicle were calculated using the profile function of Zen Blue. The RGB ratio of each vesicle was converted into a hue angle and saturation range – which we assigned to each AV subtype and should match the desired AV color range as perceived visually (i.e. yellow for AP, blue for Ly, purple for AL and white for de-acidified AL) – by entering the values of R, G, and B for a given punctum into the formula, as follows: Hue°=IF(180/PI()*ATAN2(2*R-G-B,SQRT(3)*(G-B))<0,180/PI()*ATAN2(2*R-G-B,SQRT(3) *(G-B))+360,180/PI()*ATAN2(2*R-G-B,SQRT(3)*(G-B))). Saturation percent of the hue angle was calculated by entering the values of R, G, and B for a given punctum into the following formula = (MAX(RGB)-MIN(RGB))/SUM(MAX(RGB)+MIN(RGB))*100 (http://www.workwithcolor.com/), provided lightness is less than 1, which is the usual case for our data. The Hue angle was converted to color using the Hue color wheel. The data was pooled and categorized in Excel spreadsheets.

## Ultrastructural analyses

Vibratome brain sections (50 μm) were treated with 1% osmium tetroxide in 100 mM sodium cacodylate buffer pH 7.4 for 30 min, washed in distilled water four times (10 min/wash), and then treated with 2% aqueous uranyl acetate overnight at 4 °C in the dark. Samples were then washed and sequentially dehydrated with increasing concentrations of ethanol (20, 30, 50, 70, 90, and 100%) for 30 min each, followed by three additional treatments with 100% ethanol for 20 min each. Samples were then infiltrated with increasing concentrations of Spurr's resin (25% for 1 hr, 50% for 1 hr, 75% for 1 hr, 100% for 1 hr, 100% overnight at room temperature), and then incubated overnight at 70 °C in a resin mold. For TEM ultrastructural analysis, 70 nm sections were cut using a Leica Reichert Ultracut S ultramicrotome and a Diatome diamond knife, placed onto grids and then post-stained with 2% uranyl acetate and lead citrate. Images were taken using a Ceta Camera on a ThermoFisher Talos L120C transmission electron microscope operating at 120kV.

For post-embedding IEM, ultrathin sections were placed onto carbon formvar 75 mesh nickel grids and etched using 4% sodium metaperiodate for 10 min before being washed twice in distilled water and then blocked for 1 hr. Grids were incubated in the primary antibodies at 4 °C overnight. Next day, grids underwent seven washes in 1xPBS and were then incubated in anti-mouse or anti-rabbit 10 nm gold secondary (1/50 dilution) for 1 hr. After this, the grid was washed seven times in 1x PBS and twice in distilled water. The grids were then silver enhanced for 5 min (Nanoprobes). Grids were finally post-stained with 1% uranyl acetate for 5 min followed by two washes in water and then stained with lead citrate for 5 min followed by a final two washes in distilled water. Samples were then imaged on a ThermoFisher Talos L120C operating at 120 kV.

## Western blotting

Samples for WB were prepared by homogenizing brains in a tissue-homogenizing buffer (250 mM sucrose, 20 mM Tris pH 7.4, 1 mM EDTA, 1 mM EGTA) containing protease and phosphatase inhibitors as previously described. (*Schmidt et al., 2012*) Following electrophoresis, proteins were transferred onto 0.2 μm-pore nitrocellulose membranes (Whatman, Florham Park, NJ) at 100 mA for 8–12 hr depending on the target protein. The blots were blocked for 1 hr in 5% non-fat milk in TBS, rinsed in TBST (TBS +0.1% Tween-20), then incubated with a primary antibody in 1% BSA/TBST overnight at 4 °C. The membrane was washed and incubated in a HRP-conjugated goat-anti-rabbit or mouse secondary antibody, diluted 1:5000 in 5% milk for 1 hr at room temperature. The membrane was again washed and then incubated in a Novex ECL (Invitrogen) for 1 min. The detection of the signals was achieved through either exposure to a film or scanning by a digital gel imager (Syngene G:Box XX9) as specified in the figure legends. Densitometry was performed with Image J and the results were

normalized by the immunoblot(s) of given loading control protein(s) (usually GAPDH, unless otherwise noted).

## Materials availability

Requests for resources and reagents should be directed to the lead contact, Dr. Ralph Nixon (nixon@nki.rfmh.org). The crossed TRGL/Q175 mice generated from this study are not available since the breeding had been stopped. However, Q175 mice are available from the Jackson Lab, and the TRGL mice are available from the lead contact and may require completion of a materials transfer agreement.

## Acknowledgements

We are grateful to Dr. T Yoshimori (Osaka University, Japan) for the mRFP-eEGFP-LC3 construct used in the TRGL mice. This work was supported by the CHDI Foundation (RAN) and the National Institute of Aging (P01 AG017617 to RAN).

## Additional information

### Funding

| Funder | Grant reference number | Author |
|---|---|---|
| CHDI Foundation | | Ralph A Nixon |
| National Institutes of Health | P01 AG017617 | Ralph A Nixon |

The funders had no role in study design, data collection and interpretation, or the decision to submit the work for publication.

### Author contributions

Philip Stavrides, Chris N Goulbourne, Data curation, Formal analysis, Validation, Investigation, Visualization, Methodology, Writing – review and editing; James Peddy, Data curation, Formal analysis, Investigation, Visualization; Chunfeng Huo, Data curation, Investigation; Mala Rao, Resources, Writing – review and editing; Vinod Khetarpal, Data curation, Formal analysis, Investigation, Methodology; Deanna M Marchionini, Conceptualization, Resources, Formal analysis, Validation, Methodology, Project administration, Writing – review and editing; Ralph A Nixon, Conceptualization, Resources, Supervision, Funding acquisition, Project administration, Writing – review and editing; Dun-Sheng Yang, Conceptualization, Data curation, Formal analysis, Supervision, Validation, Investigation, Visualization, Writing – original draft, Project administration, Writing – review and editing

### Author ORCIDs

Philip Stavrides ![ORCID] https://orcid.org/0000-0003-1225-3577
Mala Rao ![ORCID] https://orcid.org/0000-0001-7985-2414
Vinod Khetarpal ![ORCID] https://orcid.org/0000-0002-2175-5117
Deanna M Marchionini ![ORCID] https://orcid.org/0000-0003-2071-1064
Ralph A Nixon ![ORCID] https://orcid.org/0000-0001-5124-1021
Dun-Sheng Yang ![ORCID] https://orcid.org/0000-0001-9259-3015

### Ethics

The animal procedures conducted in this study were approved by the Institutional Animal Care and Use Committee at the NKI, AP2018-624.

Reviewer #1 (Public review): https://doi.org/10.7554/eLife.104979.3.sa1
Reviewer #2 (Public review): https://doi.org/10.7554/eLife.104979.3.sa2
Author response https://doi.org/10.7554/eLife.104979.3.sa3

## Additional files

### Supplementary files
MDAR checklist

### Data availability
No large datasets or custom codes were generated in this study. Original western blot images are uploaded as zipped source data files. Please note that in *Figure 3—figure supplements 1–4*, there are total six protein markers (MTOR, Atg5, Beclin 1, TRAF6, CTSB and CTSD) for which the presented western blot images (the lanes arranged as TRGL, TRGL/Q175, TRGL, TRGL/Q175, TRGL, TRGL/Q175) are just representative images of the signal seen but not the blots used for quantitative analysis to generate the bar graphs (where Q175 mouse samples were included in the gels, lacking the pattern of the above TRGL, TRGL/Q175, TRGL, TRGL/Q175, TRGL, TRGL/Q175). Thus, original western blots for both situations, whose file names contain either 'used for display-only' or 'used for quantitation-only', are included in the source data files.

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
