## [Editor Report · eLife Assessment]

This study presents an **important** finding on the alterations in the autophagic-lysosomal pathway in a Huntington's disease model. The evidence supporting the claims of the authors is **convincing**. The original reviewers have found most of the issues previously raised have been addressed although further suggestions are given for consideration. These comments are listed below. The work will be of interest to neuroscientists working on HD.

---

## [Referee Report · Reviewer #1 (Public review)]

Summary:

Huntington's disease (HD) is characterized by the expansion of polyglutamine repeats in huntingtin protein (HTT), leading to the formation of aggresomes composed of mutant huntingtin (mHTT). This study investigates the potential therapeutic strategy of enhancing autophagy to clear mHTT. The authors' evaluation of the autophagic-lysosomal pathway (ALP) in human HD brains shows that, in early stages, there is upregulated lysosomal biogenesis and relatively normal autophagy flux, while late-stage brains exhibit impaired autolysosome clearance, suggesting that early intervention may be beneficial. The authors cross the Q175 HD knock-in model with the TRGL autophagy reporter mouse to investigate ALP dynamics in vivo. In these models, mHTT is detected in autophagic vacuoles and colocalizes with autophagy receptors p62/SQSTM1 and ubiquitin. Although ALP alterations in the Q175 model are milder and later onset compared to human HD, they do show lysosome depletion and impaired autophagic flux. Treatment with an mTOR inhibitor in 6-month-old TRGL/Q175 mice normalized lysosome numbers, alleviated aggresome pathology, and reduced mHTT, p62, and ubiquitin levels. These findings suggest that autophagy modulation during the early stages of disease progression may offer potential therapeutic interventions for HD pathology.

Strengths:

Provide supportive animal evidence for mTOR inhibition in enhancing autophagy and reducing toxicity in HD animal models.

Weaknesses:

Lacks animal behavior and survival rate data, particularly regarding whether the extent of motor dysfunction in TRGL/Q175 mice is comparable to that in Q175 mice and whether the administration of mTORi INK improves these symptoms.

---

## [Referee Report · Reviewer #2 (Public review)]

Summary:

In this manuscript the authors have explored the beneficial effect of autophagy upregulation in the context of HD pathology in a disease stage-specific manner. The authors have observed functional autophagy lysosomal pathway (ALP) and its machineries at the early stage in HD mouse model, whereas impairment of ALP has been documented at the later stages of the disease progression. Eventually, the authors have taken advantage of operational ALP pathway at the early stage of HD pathology, in order to upregulate ALP and autophagy flux by inhibiting mTORC1 in vivo, which ultimately reverted back multiple ALP-related abnormalities and phenotypes. Therefore, this manuscript is a promising effort to shed light on the therapeutic interventions with which HD pathology can be treated at the patient level in future.

Strengths:

The study has shown alteration of ALP in HD mouse model in a very detailed manner. Such stage dependent in vivo study will be informative and has not been done before. Also, this research provides possible therapeutic intervention in patients in future.

Weaknesses:

In this revised version of the manuscript, the authors have satisfactorily addressed all the concerns raised by the reviewers. They have also provided futuristic viewpoints towards tackling neurodegenerative disorder, especially Huntington Disease (HD).

---

## [Author Response]

The following is the authors’ response to the original reviews

**Public Reviews:**

**Reviewer #1 (Public review):**
This study investigates alterations in the autophagic-lysosomal pathway in the Q175 HD knock-in model crossed with the TRGL autophagy reporter mouse. The findings provide valuable insights into autophagy dynamics in HD and the potential therapeutic benefits of modulating this pathway. The study suggests that autophagy stimulation may offer therapeutic benefits in the early stages of HD progression, with mTOR inhibition showing promise in ameliorating lysosomal pathology and reducing mutant huntingtin accumulation.However, the data raises concerns regarding the strength of the evidence. The observed changes in autophagic markers, such as autolysosome and lysosome numbers, are relatively modest, and the Western blot results do not fully match the quantitative results. These discrepancies highlight the need for further validation and more pronounced effects to strengthen the conclusions. While the study suggests the potential of autophagy regulation as a long-term therapeutic strategy, additional experiments and more reliable data are necessary to confirm the broader applicability of the TRGL/Q175 mouse model.Furthermore, the 2004 publication by Ravikumar et al. demonstrated that inhibition of mTOR by rapamycin or the rapamycin ester CCI-779 induces autophagy and reduces the toxicity of polyglutamine expansions in fly and mouse models of Huntington's disease. mTOR is a key regulator of autophagy, and its inhibition has been explored as a therapeutic strategy for various neurodegenerative diseases, including HD. Studies suggest that inhibiting mTOR enhances autophagy, leading to the clearance of mHTT aggregates. Given that dysfunction of the autophagic-lysosomal pathway and lysosomal function in HD is already well-established, and that mTOR inhibition as a therapeutic approach for HD is also known, this study does not present entirely novel findings.Major Concerns:(1) In Figure 3A1 and A2, delayed and/or deficient acidification of AL causes deficits in the reformation of LY to replenish the LY pool. However, in Figure S2D, there is no difference in AL formation or substrate degradation, as shown by the Western blotting results for CTSD and CTSB. How can these discrepancies be explained?

We appreciate the reviewer raising this point, and we agree with the concern. Please note that the material used for our immunoblotting was hemibrain homogenates, containing not only neurons but also glial cells, so the results for any protein, e.g., CTSD or CTSB in Fig. S2D, represented combined signals from neurons and glial cells. Our longstanding experience with western blot analysis of autophagy pathway markers is that signals from glial cells significantly interfere with/dilute the signals from neurons. By contrast, the immunofluorescence (IF) results in Fig. 3A, obtained with the assistance of tfLC3 probe and hue angle-based AV/LY subtype analysis, revealed the in situ conditions of the AL and LY within neurons selectively, which reflects the advantage of using the in vivo neuron-specific expression of the LC3 probe combined with IF with a LY marker in this study and our other related studies (Lee, Rao et al. 2019, Lee, Yang et al. 2022) as explained in the Introduction of this paper. Please also refer to a similar discussion regarding the WB-detected protein levels of p-ATG14 in L542-547.

(2) The results demonstrate that in the brain sections of 17-month-old TRGL/Q175 mice, there was an increase in the number of acidic autolysosomes (AL), including poorly acidified autolysosomes (pa-AL), alongside a decrease in lysosome (LY) numbers. These AL/pa-AL changes were not significant in 2-month-old or 7-month-old TRGL/Q175 mice, where only a reduction in lysosome numbers was observed. This indicates that these changes, representing damage to the autophagy-lysosome pathway (ALP), manifest only at later stages of the disease. Considering that the ALP is affected predominantly in the advanced stages of the disease (e.g., at 17 months), why were 6-month-old TRGL/Q175 mice selected for oral mTORi INK treatment, and why was the treatment duration restricted to just 3 weeks?

We thank the reviewer for the comment. A key outcome measure in our evaluation of mTORi treatment was amelioration of mHTT pathology, i.e., mHTT aggregates/IBs. Before conducting the mTORi treatment experiments, we had learned from our assessments of age-associated progression of mHTT aggresomes/IBs in mice of different ages (e.g., 2-, 6-, 10- and 17-mo) that there were already severe mHTT accumulations in Q175 at 10-mo-old (e.g., Fig. 2A). This is consistent with a previous report (Carty, Berson et al. 2015) showing that striatal mHTT inclusions dynamically increase from 4 to 8 months. From a therapeutic point of view, more aggregates in the mouse brain would make it more difficult for the autophagy machinery to clear these aggregates. Thus, the high degree of aggregates in 10- or 17-mo may not be modifiable by the mTORi and/or prevent reliable/sensitive measurements on mTORi-induced phenotype changes. We then preferred to apply the treatment to younger (i.e., 6-mo-old) mice when the mHTT pathology was not so severe, with detectable, albeit mild, ALP abnormality. Additionally, due to the 2-year funding limit for this project, there was insufficient time to generate a large set of old mice (e.g., ~18-mo) for another drug treatment experiment. In future studies, it might be worthy to conduct the treatment “in the advanced stages of the disease (e.g., ~18-mo)” to further examine the modification potential of the mTORi on the ALP as well as the HTT aggregations. As for the treatment duration, we were interested in an acute treatment schedule given that, in our dosing tests, we observed rapid responses to the treatment (e.g., target engagement) in a few days even with one dose, and that the 14-15-day treatments produced consistent responses (e.g., Fig. S3A). Long-term treatment, however, would be worthy testing in the future although our current study informs a therapeutic approach that has been suggested by others involving intermittent/pulsatile administration of mTOR inhibitors to minimize side effects of chronic long-term administration.

(3) Is the extent of motor dysfunction in TRGL/Q175 mice comparable to that in Q175 mice? Does the administration of mTORi INK improve these symptoms?

Unfortunately, we were unable to investigate motor functions experimentally with specific assays such as open field or rotarod tests in this study (partially affected by the falling of the funded research period within the COVID-19 pandemic peak periods in 2020). Based on our experience in handling the mice, we did not notice any obvious differences between Q175 and TRGL/Q175, and any improvements after the acute mTORi INK treatment.

(4) Why is eGFP expression not visible in Fig. 6A in TRGL-Veh mice? Additionally, why do normal (non-poly-Q) mice have fewer lysosomes (LY) than TRGL/Q175-INK mice? IHC results also show that CTSD levels are lower in TRGL mice compared to TRGL/Q175-INK mice. Does this suggest lysosome dysfunction in TRGL-Veh mice?

We appreciate the reviewer raising this point, which has been corrected (through slightly increasing the eGFP signal in the green channel and the merged channels equally for all genotypes), and the revised Fig. 6A is showing better eGFP signals. Regarding higher LY numbers/CTSD levels in TRGL/Q175-INK compared to the control TRGL-Veh mice, it does not necessarily imply LY dysfunction in TRGL mice, rather, it likely suggests mTORi treatment inducing LY biogenesis. Our original characterization of the TRGL mouse of varying ages, where low expression of the tgLC3 construct, produces only a very small increment of total LC3, resulting in no discernable functional changes in the autophagy pathway (Lee, Rao et al. 2019). The underlying mechanism, e.g., TFEB activation following mTOR inhibition, remains to be investigated in future studies.

(5) In Figure 5A, the phosphorylation of ATG14 (S29) shows minimal differences in Western blotting, which appears inconsistent with the quantitative results. A similar issue is observed in the quantification of Endo-LC3.

We welcome the reviewer’s point, and therefore bands showing bigger differences of p-ATG14 (S29) have been used in the revised Fig. 5A, making the images and the quantitative results more consistent and representative. Similar changes have also been made to the Endo-LC3 data at the bottom of Fig. 5A.

(6) In Figure S2A and Figure S2B, 17-month-old TRGL/Q175 mice show a decrease in pp70S6K and the p-ULK1/ULK1 ratio, but no changes are observed in autophagy-related markers. Do these results indicate only a slight change in autophagy at this stage in TRGL/Q175 mice? Since the mTOR pathway regulates multiple cellular mechanisms, could mTOR also influence other processes? Is it possible that additional mechanisms are involved?

We completely agree with the reviewer. As mentioned in the text at multiple locations, LAP alterations in Q175 and TRGL/Q175 mice are mild even at a relatively old age (e.g., 17-mo), especially at the protein levels detected by immunoblotting. We agree that even if the mild alterations in the levels of pp70S6K (T389) and p-ULK1/ULK1 ratio may indicate “a slight change in autophagy”, it may also imply that other cell processes are involved given that mTOR signaling regulates multiple cellular functions. In particular, the p70S6K/p-p70S6K – a mTOR substrate used as a readout for mTOR activity in this study – is a key component of the protein synthesis pathway (Wang and Proud 2006, Magnuson, Ekim et al. 2012) , so its changes may serve as readouts for alterations in not only the autophagy pathway, but also the protein synthesis pathway. [A related discussion about mTOR/protein synthesis pathways, in response to a comment from Reviewer 2, has been incorporated into the text under Discussion, L633-640]

**Reviewer #2 (Public review):**
Summary:In this manuscript, the authors have explored the beneficial effect of autophagy upregulation in the context of HD pathology in a disease stage-specific manner. The authors have observed functional autophagy lysosomal pathway (ALP) and its machineries at the early stage in the HD mouse model, whereas impairment of ALP has been documented at the later stages of the disease progression. Eventually, the authors took advantage of the operational ALP pathway at the early stage of HD pathology, in order to upregulate ALP and autophagy flux by inhibiting mTORC1 in vivo, which ultimately reverted back to multiple ALP-related abnormalities and phenotypes. Therefore, this manuscript is a promising effort to shed light on the therapeutic interventions with which HD pathology can be treated at the patient level in the future.Strengths:The study has shown the alteration of ALP in the HD mouse model in a very detailed manner. Such stage-dependent in vivo study will be informative and has not been done before. Also, this research provides possible therapeutic interventions for patients in the future.Weaknesses:Some constructive comments and suggestions in order to reflect the key aspects and concepts better in the manuscript :(1) The authors have observed lysosome number alteration in a temporally regulated disease stage-specific manner. In this scenario investigation of regulation, localization, and level of TFEB, the transcription factor required for lysosome biogenesis, would be interesting and informative.

We thank the reviewer for this point and completely agree that exploring TFEBrelated aspects would be interesting which will be investigated in future studies.

(2) For the general scientific community better clarification of the short forms will be useful. For example, in line 97, page 4, AP full form would be useful. Also 'metabolized via autophagy' can be replaced by 'degraded via autophagy'.

We appreciate the reviewer for raising this point. We introduced each abbreviation at the location where the full term first appears and, for the case of “AP”, it was introduced in (previous) Line 69 when “autophagosome” first appears. We agree with the reviewer about easy reading for the general scientific community and thus we have added an Abbreviation section after the Key Words section, listing abbreviations used in this manuscript.

Also, the word “metabolized” has been replaced with “degraded” as suggested.

(3) The nuclear vs cytosolic localization of HTT aggregates shown in Figure 2, are very interesting. The increase in cytosolic HTT aggregate formation at 10 months compared to 6 months probably suggests spatio-temporal regulation of aggregate formation. The authors could comment in a more elaborate manner, on the reason and impact of this kind of regulation of aggregate formation in the context of HD pathology.

We value the reviewer’s important point. Previous studies have well documented that mHTT aggregates exist in both intranuclear and extranuclear locations in the brains of both human HD and mouse models (DiFiglia, Sapp et al. 1997, Li, Li et al. 1999, Carty, Berson et al. 2015, Peng, Wu et al. 2016, Berg, Veeranna et al. 2024). HTT can travel between the nucleus and cytoplasm and the default location for HTT is cytoplasmic, and thus the occurrence of nuclear mHTT aggregates is considered as a result of dysfunction in the nuclear exporting system for proteins (DiFiglia, Sapp et al. 1995, Gutekunst, Levey et al. 1995, Sharp, Loev et al. 1995, Cornett, Cao et al. 2005) while other factors such as phosphorylation of HTT may also affect nuclear targeting (DeGuire, Ruggeri et al. 2018). Extranuclear aggregates of mHTT usually appear later than nuclear aggregates and develop more aggressively in terms of numbers and pace after their appearance (Li, Li et al. 1999, Carty, Berson et al. 2015, Landles, Milton et al. 2020). The fact that there are neurons containing extranuclear aggregates without having nuclear aggregates within the same cells (Carty, Berson et al. 2015) does not support a nuclear-cytoplasmic sequence for aggregate formation, implying different mechanisms controlling the formation of these two types of aggregates. It was reported that there were no significant differences in toxicity associated with the presence of nuclear compared with extranuclear aggregates (Hackam, Singaraja et al. 1999), while other studies have proposed that nuclear aggregates correlate with transcriptional dysfunction while extranuclear aggregates may impair neuronal communication and can track disease progression (Li, Li et al. 1999, Benn, Landles et al. 2005, Landles, Milton et al. 2020). Thus, the observation of a higher level of extranuclear mHTT aggregates at 10-mo compared to 6-mo from the present study is consistent with previous findings mentioned above. In addition, our EM observations of homogenous granular/short fine fibril ultrastructure of both nuclear and extranuclear aggregates are consistent with findings from mouse model studies (Davies, Turmaine et al. 1997, Scherzinger, Lurz et al. 1997), which, interestingly, is different from in vitro studies where nuclear aggregates exhibited a core and shell structure but extranuclear aggregates did not possess the shell (Riguet, Mahul-Mellier et al. 2021), reflecting differences between in vivo and in vitro conditions. Taken together, even if efforts have been made in this and previous studies in trying to understand the differences between nuclear and extranuclear aggregates, the mechanisms regarding the spatial-temporal regulation of aggregate formation have so far not been fully revealed which will require additional investigations.

(4) In this manuscript, the authors have convincingly shown that mTOR inhibition is inducing autophagy in the HD mouse model in vivo. On the other hand, mTOR inhibition would also reduce overall cellular protein translation. This aspect of mTOR inhibition can also potentially contribute to the alleviation of disease phenotype and disease symptoms by reducing protein overload in HD pathology. The authors' comments regarding this aspect would be appreciated.

We recognize the value of the reviewer’s point which we completely agree with. Lowering mHTT via interfering protein translation (e.g., through RNAi, antisense oligonucleotides) has been an attractive strategy in HD therapeutic development (Kordasiewicz, Stanek et al. 2012, Tabrizi, Ghosh et al. 2019). As mentioned above, mTOR regulates multiple cellular pathways including protein synthesis, and inhibition of mTOR as what was done in the present study is potentially affect protein synthesis as well. While our results of decreases in mHTT signals (Fig. 7) can be interpreted as a result of autophagymediated clearance of mHTT, certainly, a possibility cannot be excluded that mTOR inhibition may result in a reduction in HTT production which may also contribute to the observed results – future studies should determine how significant of such a contribution is. [The above description has been incorporated into the text under Discussion, L633-640]

(5) The authors have shown nuclear inclusion formation and aggregation of mHTT and also commented on its potential removal with the UPS system (proteasomal degradation) in vivo. As there is also a reciprocal relationship present between autophagy and proteasomal machineries, upon upregulation of autophagy machinery by mTOR inhibition proteasomal activity may decrease. How nuclear proteasomal activity increases to tackle nuclear mHTT IBs, would be interesting to understand in the context of HD pathology. Comments from the authors in this aspect would clarify the role of multiple degradation pathways in handling mutant HTT protein in HD pathology.

We appreciate the reviewer raising this point. We agree that there are reciprocal relationships between autophagy and the UPS (Korolchuk, Menzies et al. 2010, Park and Cuervo 2013). In general, failure in one pathway would lead to compensatory upregulation of the other pathway, and vice versa (Lee, Park et al. 2019). So, as the reviewer pointed out, “upon upregulation of autophagy machinery by mTOR inhibition proteasomal activity may decrease”. However, we proposed in the Discussion that “It is possible that stimulation of autophagy is reducing the mHTT in the cytoplasm and thereby partially relieves the burden of the proteasome both in the cytoplasm and in the nucleus so that the nuclear proteasome operates more effectively”, which is inconsistent with the general expectation for a decreased UPS activity. However, please note that there are also instances where two pathways may act in the same direction, e.g., autophagy inhibition disturbs UPS degradative function (Korolchuk, Mansilla et al. 2009, Park and Cuervo 2013). Anyhow, our statement is just speculation, requiring verifications with additional experiments in the future. One of the observations reported here which may support the above speculation is the reductions of AV-non-associated form of mHTT/p62/Ub (Fig. 7B3), given that some of them might exist within the nucleus, whose reduced levels may reflect increased intranuclear UPS activity, besides the other possibility that they may travel from the nucleus to the cytosol for clearance as already discussed inside the text. [The last sentence has been incorporated into the text under Discussion, L628-632]

(6) For the treatment of neurodegenerative disorders taking the temporal regulation into consideration is extremely important, as that will determine the success rate of the treatments in patients. The authors in this manuscript have clearly discussed this scenario. However, for neurodegenerative disordered patients, in most cases, the symptom manifestation is a late onset scenario. In that case, it will be complicated to initiate an early treatment regime in HD patients. If the authors can comment on and discuss the practicality of the early treatment regime for therapeutic purposes that would be impactful.

We appreciate the reviewer raising this point and we agree with the main concern that “for neurodegenerative disordered patients, in most cases, the symptom manifestation is a late onset scenario.” This is really a common challenge in the therapeutic fields for neurodegeneration diseases. It should be first noted that the current study is an experimental therapeutical attempt in a mouse model which is consistent with previous reports (Ravikumar, Vacher et al. 2004) as a proof of concept for manipulating autophagy (i.e., via inhibiting mTOR in the current setting) as a potential therapeutic, whose clinical practicality requires further verifications. Moreover, in our opinion, early diagnosis (e.g., genetic testing in individuals with higher risk for HD) may be a key in overcoming the above challenges, i.e., if early diagnosis is enabled, it would become possible for earlier interventions. [The above description has been incorporated into the text under Discussion, L654-659]

**Recommendations for the authors:**
Reviewer #1 (Recommendations for the authors):Minor concerns:(1) Figures 1 and 2 should indicate the number of sections and mice/genotypes.

Thanks for the suggestion, and the info has been added in the figure legends.

(2) Figure 3A2 should explain how AP, AL, pa-AL, and LY are quantified.

Thanks for raising this point. Please note that the quantitation of AP, AL, pa-AL and LY was performed by the hue angle-based analysis which was described under “Confocal image collection and hue angle-based quantitative analysis for AV/LY subtypes” within the Materials and Methods. A phrase “(see the Materials and Methods)” has been added after the existing description “Hue angle-based analysis was performed for AV/LY subtype determination using the methods described in Lee et al., 2019” in the figure legend.

References

Benn, C. L., C. Landles, H. Li, A. D. Strand, B. Woodman, K. Sathasivam, S. H. Li, S. Ghazi-Noori, E. Hockly, S. M. Faruque, J. H. Cha, P. T. Sharpe, J. M. Olson, X. J. Li and G. P. Bates (2005). "Contribution of nuclear and extranuclear polyQ to neurological phenotypes in mouse models of Huntington's disease." Hum Mol Genet 14(20): 3065-3078.

Berg, M. J., Veeranna, C. M. Rosa, A. Kumar, P. S. Mohan, P. Stavrides, D. M. Marchionini, D.S. Yang and R. A. Nixon (2024). "Pathobiology of the autophagy-lysosomal pathway in the Huntington’s disease brain." bioRxiv: 2024.2005.2029.596470.

Carty, N., N. Berson, K. Tillack, C. Thiede, D. Scholz, K. Kottig, Y. Sedaghat, C. Gabrysiak, G. Yohrling, H. von der Kammer, A. Ebneth, V. Mack, I. Munoz-Sanjuan and S. Kwak (2015). "Characterization of HTT inclusion size, location, and timing in the zQ175 mouse model of Huntington's disease: an in vivo high-content imaging study." PLoS One 10(4): e0123527.

Cornett, J., F. Cao, C. E. Wang, C. A. Ross, G. P. Bates, S. H. Li and X. J. Li (2005). "Polyglutamine expansion of huntingtin impairs its nuclear export." Nat Genet 37(2): 198204.

Davies, S. W., M. Turmaine, B. A. Cozens, M. DiFiglia, A. H. Sharp, C. A. Ross, E. Scherzinger, E. E. Wanker, L. Mangiarini and G. P. Bates (1997). "Formation of neuronal intranuclear inclusions underlies the neurological dysfunction in mice transgenic for the HD mutation." Cell 90(3): 537-548.

DeGuire, S. M., F. S. Ruggeri, M. B. Fares, A. Chiki, U. Cendrowska, G. Dietler and H. A. Lashuel (2018). "N-terminal Huntingtin (Htt) phosphorylation is a molecular switch regulating Htt aggregation, helical conformation, internalization, and nuclear targeting." J Biol Chem 293(48): 18540-18558.

DiFiglia, M., E. Sapp, K. Chase, C. Schwarz, A. Meloni, C. Young, E. Martin, J. P. Vonsattel, R. Carraway, S. A. Reeves and et al. (1995). "Huntingtin is a cytoplasmic protein associated with vesicles in human and rat brain neurons." Neuron 14(5): 1075-1081.

DiFiglia, M., E. Sapp, K. O. Chase, S. W. Davies, G. P. Bates, J. P. Vonsattel and N. Aronin (1997). "Aggregation of huntingtin in neuronal intranuclear inclusions and dystrophic neurites in brain." Science 277(5334): 1990-1993.

Gutekunst, C. A., A. I. Levey, C. J. Heilman, W. L. Whaley, H. Yi, N. R. Nash, H. D. Rees, J. J. Madden and S. M. Hersch (1995). "Identification and localization of huntingtin in brain and human lymphoblastoid cell lines with anti-fusion protein antibodies." Proc Natl Acad Sci U S A 92(19): 8710-8714.

Hackam, A. S., R. Singaraja, T. Zhang, L. Gan and M. R. Hayden (1999). "In vitro evidence for both the nucleus and cytoplasm as subcellular sites of pathogenesis in Huntington's disease." Hum Mol Genet 8(1): 25-33.

Kordasiewicz, H. B., L. M. Stanek, E. V. Wancewicz, C. Mazur, M. M. McAlonis, K. A. Pytel, J. W. Artates, A. Weiss, S. H. Cheng, L. S. Shihabuddin, G. Hung, C. F. Bennett and D. W. Cleveland (2012). "Sustained therapeutic reversal of Huntington's disease by transient repression of huntingtin synthesis." Neuron 74(6): 1031-1044.

Korolchuk, V. I., A. Mansilla, F. M. Menzies and D. C. Rubinsztein (2009). "Autophagy inhibition compromises degradation of ubiquitin-proteasome pathway substrates." Mol Cell 33(4): 517-527.

Korolchuk, V. I., F. M. Menzies and D. C. Rubinsztein (2010). "Mechanisms of cross-talk between the ubiquitin-proteasome and autophagy-lysosome systems." FEBS Lett 584(7): 1393-1398.

Landles, C., R. E. Milton, N. Ali, R. Flomen, M. Flower, F. Schindler, C. Gomez-Paredes, M. K. Bondulich, G. F. Osborne, D. Goodwin, G. Salsbury, C. L. Benn, K. Sathasivam, E. J. Smith, S. J. Tabrizi, E. E. Wanker and G. P. Bates (2020). "Subcellular Localization And Formation Of Huntingtin Aggregates Correlates With Symptom Onset And Progression In A Huntington'S Disease Model." Brain Commun 2(2): fcaa066.

Lee, J. H., S. Park, E. Kim and M. J. Lee (2019). "Negative-feedback coordination between proteasomal activity and autophagic flux." Autophagy 15(4): 726-728.

Lee, J. H., M. V. Rao, D. S. Yang, P. Stavrides, E. Im, A. Pensalfini, C. Huo, P. Sarkar, T. Yoshimori and R. A. Nixon (2019). "Transgenic expression of a ratiometric autophagy probe specifically in neurons enables the interrogation of brain autophagy in vivo." Autophagy 15(3): 543-557.

Lee, J. H., D. S. Yang, C. N. Goulbourne, E. Im, P. Stavrides, A. Pensalfini, H. Chan, C. Bouchet-Marquis, C. Bleiwas, M. J. Berg, C. Huo, J. Peddy, M. Pawlik, E. Levy, M. Rao, M. Staufenbiel and R. A. Nixon (2022). "Faulty autolysosome acidification in Alzheimer's disease mouse models induces autophagic build-up of Abeta in neurons, yielding senile plaques." Nat Neurosci 25(6): 688-701.

Li, H., S. H. Li, A. L. Cheng, L. Mangiarini, G. P. Bates and X. J. Li (1999). "Ultrastructural localization and progressive formation of neuropil aggregates in Huntington's disease transgenic mice." Hum Mol Genet 8(7): 1227-1236.

Magnuson, B., B. Ekim and D. C. Fingar (2012). "Regulation and function of ribosomal protein S6 kinase (S6K) within mTOR signalling networks." Biochem J 441(1): 1-21.

Park, C. and A. M. Cuervo (2013). "Selective autophagy: talking with the UPS." Cell Biochem Biophys 67(1): 3-13.

Peng, Q., B. Wu, M. Jiang, J. Jin, Z. Hou, J. Zheng, J. Zhang and W. Duan (2016). "Characterization of Behavioral, Neuropathological, Brain Metabolic and Key Molecular Changes in zQ175 Knock-In Mouse Model of Huntington's Disease." PLoS One 11(2): e0148839.

Ravikumar, B., C. Vacher, Z. Berger, J. E. Davies, S. Luo, L. G. Oroz, F. Scaravilli, D. F. Easton, R. Duden, C. J. O'Kane and D. C. Rubinsztein (2004). "Inhibition of mTOR induces autophagy and reduces toxicity of polyglutamine expansions in fly and mouse models of Huntington disease." Nat Genet 36(6): 585-595.

Riguet, N., A. L. Mahul-Mellier, N. Maharjan, J. Burtscher, M. Croisier, G. Knott, J. Hastings, A. Patin, V. Reiterer, H. Farhan, S. Nasarov and H. A. Lashuel (2021). "Nuclear and cytoplasmic huntingtin inclusions exhibit distinct biochemical composition, interactome and ultrastructural properties." Nat Commun 12(1): 6579.

Scherzinger, E., R. Lurz, M. Turmaine, L. Mangiarini, B. Hollenbach, R. Hasenbank, G. P. Bates, S. W. Davies, H. Lehrach and E. E. Wanker (1997). "Huntingtin-encoded polyglutamine expansions form amyloid-like protein aggregates in vitro and in vivo." Cell 90(3): 549-558.

Sharp, A. H., S. J. Loev, G. Schilling, S. H. Li, X. J. Li, J. Bao, M. V. Wagster, J. A. Kotzuk, J. P. Steiner, A. Lo and et al. (1995). "Widespread expression of Huntington's disease gene (IT15) protein product." Neuron 14(5): 1065-1074.

Tabrizi, S. J., R. Ghosh and B. R. Leavitt (2019). "Huntingtin Lowering Strategies for Disease Modification in Huntington's Disease." Neuron 101(5): 801-819.

Wang, X. and C. G. Proud (2006). "The mTOR pathway in the control of protein synthesis." Physiology (Bethesda) 21: 362-369.